# FlashRNN: I/O-Aware Optimization of Traditional RNNs on modern hardware

**Korbinian Pöppel & Maximilian Beck & Sepp Hochreiter**
Johannes Kepler University, NXAI Lab and NXAI GmbH
Altenberger Str. 69
A-4040 Linz, Austria
{poeppel,beck,hochreit}@ml.jku.at

## Abstract

While Transformers and other sequence-parallelizable neural network architectures seem like the current state of the art in sequence modeling, they specifically lack state-tracking capabilities. These are important for time-series tasks and logical reasoning. Traditional RNNs like LSTMs and GRUs, as well as modern variants like sLSTM do have these capabilities at the cost of strictly sequential processing. While this is often seen as a strong limitation, we show how fast these networks can get with our hardware-optimization FlashRNN in Triton and CUDA, optimizing kernels to the register level on modern GPUs. We extend traditional RNNs with a parallelization variant that processes multiple RNNs of smaller hidden state in parallel, similar to the head-wise processing in Transformers. To enable flexibility on different GPU variants, we introduce a new optimization framework for hardware-internal cache sizes, memory and compute handling. It models the hardware in a setting using polyhedral-like constraints, including the notion of divisibility. This speeds up the solution process in our ConstrINT library for general integer constraint satisfaction problems (integer CSPs). We show that our kernels can achieve 50x speed-ups over a vanilla PyTorch implementation and allow 40x larger hidden sizes compared to our Triton implementation. We have open-sourced our kernels and the optimization library to boost research in the direction of state-tracking enabled RNNs and sequence modeling here: https://github.com/NX-AI/flashrnn.

## 1 Introduction

Sequence models are at the core of many applications like time-series modeling, natural language processing, text, audio and video models, and predictions for physical systems based on ODEs or PDEs (Vaswani et al., 2017; Degrave et al., 2022; Nearing et al., 2024). While there are modern sequence-parallelizable architectures like the Transformer (Vaswani et al., 2017), Mamba (Gu & Dao, 2023) or mLSTM (Beck et al., 2024), these lack the state-tracking capabilities (Merrill et al., 2024; Merrill & Sabharwal, 2023; Delétang et al., 2023) of traditional RNNs like LSTM (Hochreiter & Schmidhuber, 1997), GRU (Cho et al., 2014) and sLSTM (Beck et al., 2024).

Traditional RNNs include a recurrent connection or memory mixing, that connects the previous hidden state in a non-linear way to the current state update and this way mixes the states of different memory cells. While the sequence has to be processed step by step, computed hidden states and the recurrent matrix weights can stay cached, enabling a large speed optimization. In this work, we introduce FlashRNN as a generic hardware-optimized library for these RNN-style architectures.

Our library facilitates research in the direction of state-tracking enabled RNN architectures, in two ways: Firstly, it enables easier and more efficient use of recent RNN-architectures like sLSTM (Beck et al., 2024). This includes the notion of block-diagonal recurrent matrices that can speed up networks while lowering the number of parameters. Secondly, it can be easily extended to novel RNN-like architecture variants, as it supports generic state and gate numbers per cell. The LSTM (Hochreiter & Schmidhuber, 1997; Gers et al., 1999), with its two states and four gates (we consider the cell update as a fourth "gate" for simplicity here), can be implemented as easy as a simple Elman-RNN

with one state and one gate (Elman, 1990), or sLSTM with its three states and four gates (Beck et al., 2024).

To realize the shown speed-ups, we fuse the recurrent matrix-multiplication part with the point-wise activation part, both wrapped in the sequential loop into one kernel. This can be used on different GPUs and with different state/gate variants, as our library optimizes internal memory sizes and operations automatically based on the models' hidden sizes and the cache and register sizes of the hardware.

For the auto-optimization we introduce an integer constraint satisfaction library ConstrINT. With this library, one can model generic integer CSP problems with equality, inequality and divisibility constraints as these can model size constraints on modern hardware with specific tensor-core, register and SRAM memory sizes.

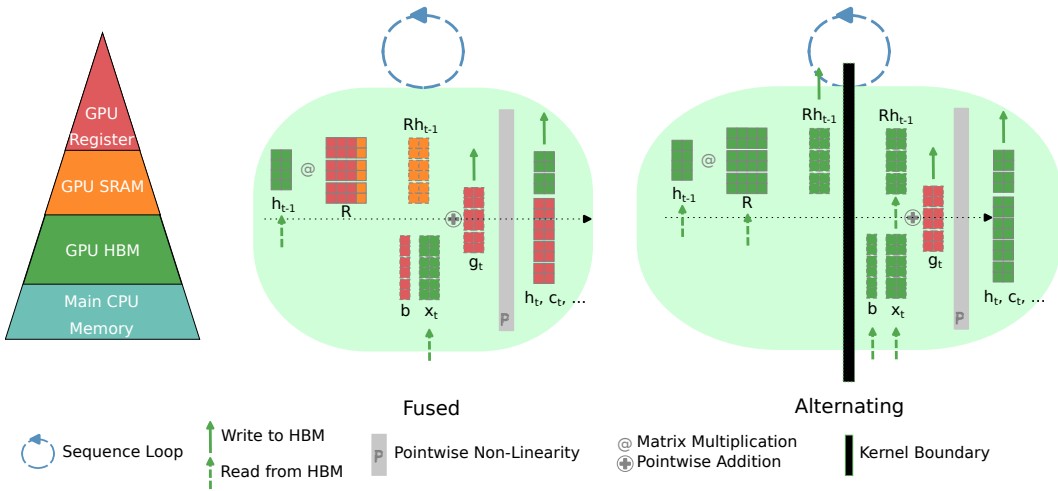

Figure 1: FlashRNN Kernel overview: Left: Basic Memory Hierarchy in modern GPUs. Center: Fused Kernel (forward) leveraging all caching options for maximal speed. Right: Alternating Kernels (forward) for maximum hidden sizes, with two kernel calls per time step. The colors show the caching level of the different tensors, the batch dimension is depicted to the right (except for R), the hidden / gate dimension vertically.

## 2 RELATED WORK

Hardware-aware algorithms and their open-source implementations of common sequence modeling primitives have been focused primarily around the Transformer architecture (Vaswani et al., 2017) and its attention operation because of its ubiquity in language modeling. FlashAttention (Dao et al., 2022) introduced an IO-aware attention algorithm and CUDA implementation that uses tiling to reduce the number of memory reads/writes between GPU high bandwidth memory (HBM) and GPU on-chip SRAM, and achieves significant memory savings. FlashAttention2 (Dao, 2024) improves FlashAttention with better work partitioning and the additional parallelization over the sequence dimension. FlashAttention3 (Shah et al., 2024) takes advantage of new capabilites, such as asynchrony and FP8 low precision support of the recent NVIDA Hopper GPU generation.

Recently, novel sequence models taking inspiration of Linear Attention (Katharopoulos et al., 2020) have shown promising performance compared to Transformer Attention (Beck et al., 2024; Yang et al., 2024; Dao & Gu, 2024). Yang et al. (2024) provide an hardware-efficient algorithm and implementation in Triton for Gated Linear Attention that trades off memory movement against parallelizability and show that it is faster than FlashAttention2.

Traditional RNNs like LSTMs (Hochreiter & Schmidhuber, 1997) or GRUs (Cho et al., 2014) are still widely used in many applications, such as for example time series modeling or reinforcement learning (Nearing et al., 2024; Degrave et al., 2022). Many of these applications rely on optimized

closed-source implementations of these RNN operations such as in the NVIDIA cuDNN [1] library, which is integrated in PyTorch. Sharvil (2020) provide an open-source alternative in CUDA for specific LSTM and GRU variants in their `HASTE` library, which served as inspiration for this work. `HASTE` is limited in speed due to a sequence of alternating calls of matrix multiplication and point-wise kernels, as well as its limitation to higher (but slower) precision.

Our work FlashRNN overcomes this limitation by fusing the recurrent matrix multiplication with the pointwise operations into a single persistent kernel with custom caching of the recurrent weights in registers. FlashRNN also supports the bfloat16 dtype and block-diagonal recurrent matrices. By open-sourcing our CUDA and Triton kernels we aim to enable researchers to quickly reach similar speeds compared to optimized closed source libraries.

# 3 GENERIC RECURRENT NEURAL NETWORK ARCHITECTURE WITH MEMORY MIXING

A generic RNN architecture that we aim to optimize has $N_s$ states $\boldsymbol{s}^{(i)} \in \mathbb{R}^d$, and $N_g$ gates (or pre-activations) $\boldsymbol{g}^{(j)} \in \mathbb{R}^d$, with $d$ being the embedding dimension or hidden size of the RNN. For example the LSTM (Hochreiter & Schmidhuber, 1997) has $N_s = 2$ states and $N_g = 4$ gates.

Each gate receives an input $\boldsymbol{x}^{(j)} \in \mathbb{R}^d$. As learnable parameters, the gates have a recurrent matrix $\boldsymbol{R}^{(j)} \in \mathbb{R}^{d \times d}$ that models the dependency on the previous hidden state $\boldsymbol{s}_{t-1}^{(0)}$ and a bias $\boldsymbol{b}^{(j)} \in \mathbb{R}^d$. The state sequence of the RNN is then defined as:

$$\boldsymbol{g}_t^{(j)} = \boldsymbol{x}_t^{(j)} + \boldsymbol{R}^{(j)} \boldsymbol{s}_{t-1}^{(0)} + \boldsymbol{b}^{(j)}, \tag{1}$$

$$\boldsymbol{s}_t^{(i)} = \mathcal{P}^{(i)} \left( \left( \boldsymbol{s}_{t-1}^{(i')} \right)_{i' \in \{1..N_s\}}, \left( \boldsymbol{g}_t^{(j)} \right)_{j \in \{1..N_g\}} \right), \tag{2}$$

with a point-wise / element-wise function $\mathcal{P}^{(i)}$ that does not mix different cells along the vector dimension (unlike the recurrent weight). In Appendix A, we show how this generic formulation translates to the most common RNN variants.

Usually for these networks, the input is modified with another weight matrix $\boldsymbol{W}$. We omit this here as it can be moved outside of the basic kernels. In the common training setting, where the whole sequence is given as input, the weight matrix $\boldsymbol{W}$ can be applied in parallel to all timesteps before processing a sequence in the RNN. Our runtime experiments in Section 6.1 show that this operation has only marginal impact on the overall runtime.

# 4 GENERIC GRADIENT FOR BACK-PROPAGATION THROUGH TIME

In back-propagation through time (Mozer, 1995), the backward pass of this RNN architecture can be recursively defined as well. The backward pass reads:

$$\delta \boldsymbol{g}_t^{(j)} = \frac{\partial \mathcal{P}^{(l)} \left( \left( \boldsymbol{s}_{t-1}^{(k)} \right)_{k \in \{1..N_s\}}, \left( \boldsymbol{g}_t^{(j)} \right)_{j \in \{1..N_g\}} \right)}{\partial \boldsymbol{g}_t^{(j)}} \delta \boldsymbol{s}_t^{(l)} \tag{3}$$

$$\delta \boldsymbol{s}_{t-1}^{(i)} = \frac{\partial \mathcal{P}^{(l)} \left( \left( \boldsymbol{s}_{t-1}^{(k)} \right)_{k \in \{1..N_s\}}, \left( \boldsymbol{g}_t^{(j)} \right)_{j \in \{1..N_g\}} \right)}{\partial \boldsymbol{s}_{t-1}^{(i)}} \delta \boldsymbol{s}_t^{(l)} + \left( \sum_{j \in \{1..N_g\}} \boldsymbol{R}^{(j)^T} \delta \boldsymbol{g}_{t-1}^{(j)} \quad \text{if } i = 0 \right) \tag{4}$$

The structure of the gradient shows that, also for the backward pass, we have an alternation of point-wise operations (left) and matrix multiplication (right).

The input gradient is equal to the gate gradients, the bias gradient is the sum of the input gradients and the recurrent weight matrix gradient is the time-wise sum of the outer product of gate gradients

---

[1] https://developer.nvidia.com/cudnn

with the state values:

$$\delta \boldsymbol{x}_t^{(j)} = \delta \boldsymbol{g}_t^{(j)} \tag{5}$$

$$\delta \boldsymbol{b}^{(j)} = \sum_t \delta \boldsymbol{g}_t^{(j)} \tag{6}$$

$$\delta \boldsymbol{R}^{(j)} = \sum_t \delta \boldsymbol{g}_t^{(j)} \boldsymbol{s}_t^{(0)T} \tag{7}$$

### 4.1 VANISHING AND EXPLODING GRADIENTS AND GRADIENT MODIFICATIONS

For a neural network to be stably trainable, there must not be exploding gradients, also vanishing gradients should be prohibited for long context sequence modeling (Hochreiter & Schmidhuber, 1997). Still, for the generic structure of Equations 3, there can be exploding components: Firstly, one or more eigenvalues of the point-wise function Jacobian can be greater than one in magnitude. This can be mitigated by a proper choice of the point-wise function. Secondly, the combination of recurrent matrix and gate gradients with the gradient $\frac{\partial \mathcal{P}^{(0)}}{\partial \boldsymbol{g}_t^{(j)}} \boldsymbol{R}^{(j)T}$ could have singular values of magnitude $> 1$. This case cannot be excluded directly, as the recurrent matrix consists of trainable, typically unconstrained weights. However, for practical training this is rarely a limitation.

In our library, we implement a simple approach for mitigating this at the cost of additional gradient noise, clipping the gradient values on a scalar level after each time step. Specifically, we clip the term containing the recurrent matrix to within a pre-defined magnitude. The gradients can even be cut to zero, leading to typically worse convergence at the benefit of faster training, as the recurrent matrix part in Equation 3 is cut to zero for the backward pass.

### 4.2 HEAD-WISE PARALLELIZATION

When increasing the size of a neural network, typically the width, i.e. the embedding dimension or hidden size is increased. Vaswani et al. (2017) found that for the attention operation it is beneficial to linearly project the input embedding vectors into multiple smaller input vectors, the so called heads, and then perform attention on each of these small vectors in parallel. This parallelization primitive enables also efficient implementations on GPUs, since each head can be computed in different thread blocks of the GPU (Dao et al., 2022) in parallel (see also Section 5.1).

Many more recent architectures also rely on this head-wise parallelization primitive (Beck et al., 2024; Yang et al., 2024; Dao & Gu, 2024), where the embedding or hidden vector of dimension $d$ is split into $N_{heads}$ heads of smaller dimension $d_{head} = d/N_{head}$, each of which is processed independently inside the sequential part. In FlashRNN, we apply this primitive to traditional RNNs by dividing the recurrent matrix $\boldsymbol{R}$ into multiple blocks or heads $\boldsymbol{R}_{head} \in \mathbb{R}^{d_{head} \times d_{head}}$ rendering the recurrent matrix $\boldsymbol{R}$ as a block-diagonal matrix.

## 5 HARDWARE-EFFICIENT IMPLEMENTATION

### 5.1 GPU-ACCLERATED COMPUTING

Modern compute hardware in the form of GPUs enables massive parallelization and accelerated matrix multiplication. This means that both point-wise (scalar) operations can be parallelized and matrix multiplications have good support via BLAS-like libraries (Lawson et al., 1979; Thakkar et al., 2023), as used for RNN training workloads as defined above.

**Execution Model**  Specifically, a modern GPU consists of larger computational super-units (i.e. streaming multiprocessors (SMs)) that have some faster memory attached to them. There are three levels of memory, the large HBM that allows global random access from all computational units at the cost of low speed (still fast compared to CPU RAM access), the SRAM that is attached to one computational super-unit and the registers that are tied to a smallest computational unit (i.e. thread). One super-unit usually supports up to 1024 threads in parallel (with varying register sizes), which are typically referred to as a block or thread block. Multiple blocks executed in parallel on mul-

tiple super-units are called the grid. [2] An NVIDIA H100, for example, consists of 132 streaming multiprocessors, with 256 KB SRAM per SM and a SRAM bandwidth of around 33 TB/s (Spector et al., 2024), compared to the up to 3 TB/s for access to the 80 GB of HBM. Starting from the NVIDIA Ampere Architecture and newer, there is hardware acceleration for asynchronous loading and SRAM interconnection, which we did not utilize in this work. [3] Beyond the memory levels, a computational super-unit allows for hardware-accelerated matrix multiplication (e.g. via Tensor-Cores, "wmma" operation). Typically, it is divided into sub-units (warps) of a certain number of threads (NVIDIA: 32) that act as one for a matrix multiplication. There are certain size limitations for this acceleration, which have to be considered in the kernel optimization process. For a NVIDIA H100, this means that only minimal matrices of sizes 32x16x8, 16x16x16 or 8x16x32 can be multiplied for the low-precision bfloat16 or float16 dtypes, larger matrix multiplications have to be composed of those, by parallelization along the outer dimensions and summation along the accumulating dimension.

**Performance measures**   The specific limitation of a computational load falls into two regimes: Being compute-bound or being memory-bound. In the former case, the arithmetic intensity is high, there are many compute operations per loaded byte and therefore, the main limitation is the computational part. In the latter case, arithmetic intensity is low and the bottleneck is the memory access to load inputs and store outputs (Williams et al., 2009). Small operations, like applying an activation function in parallel are typically memory bound and should be grouped together into a fused kernel.

**Fused Kernels**   To minimize HBM memory accesses, one combines multiple arithmetic operations in one GPU kernel. A kernel is a set of instructions on the GPU. which is executed in parallel on its parts. Only within the execution of one kernel SRAM and registers are kept and can serve as a cache. Therefore, for memory-bound operations it is helpful to fuse multiple arithmetic operations into one kernel to leverage these lower cache levels. While compilers can fuse point-wise operations, an alternation of both point-wise computations and matrix multiplication is non-trivial.

## 5.2   FLASHRNN KERNELS

As the RNN operations of Equations 1 and 3 are a sequential alternation between matrix multiplication and pointwise non-linearities, there is a simple speed up variant that optimizes these two primitives separately. Our library implements this variant, in the **alternating backend**. This enables arbitrarily large head dimensions (to the limits of HBM GPU memory). Also, a vanilla PyTorch implementation relying on auto-grads will work in this primitive, but for every time step a separate state is saved for the backward pass, leading to inefficiencies beyond memory accesses. Moving the time-loop into CUDA can already give large speedups over the vanilla PyTorch implementation.

The downside of the **alternating** implementation is that there are no I/O optimizations beyond a single time step. For every time step, the current input and last state, as well as the recurrent matrix and the biases have to be re-loaded. However, both the recurrent matrix $R$ and the biases remain the same for the whole time loop and the previous states can stay in memory as they were computed in the previous time step. Since the structure of the computation remains the same over the time steps, one can even store most of these values in registers. Registers have the highest memory bandwidth and, while they can only be used within the lowest computation unit (threads), their total size on a GPU is comparable to the SRAM (both 256 KB per SM on H100).

To reach the maximum speed, we implement FlashRNN **fused** kernels that store the recurrent matrix $R$ and the biases $b$ in registers (and SRAM if register memory is exceeded). The matrix multiplication results are stored and accumulated in shared memory (or HBM if SRAM sizes are exceeded). In the forward pass, the computations are mainly tiled along the gate dimension (or the dimension of the new hidden states). This way, we use the maximum amount of memory along the previous state dimension. This dimension is the accumulating dimension of the recurrent matrix multiplication. For the backward pass, the computations are typically tiled along the previous state gradient dimension, such that the gate dimension, which is accumulated over, is minimally tiled. Algorithm 1 shows a simplified representation of the forward pass in pseudo-code and in Appendix Section B, this algorithm is shown in more detail.

---

[2]https://docs.nvidia.com/cuda/pdf/CUDA_C_Programming_Guide.pdf
[3]https://resources.nvidia.com/en-us-tensor-core/gtc22-whitepaper-hopper

---

**Algorithm 1** FlashRNN-fused forward pass

---

All states are tiled along threads (single ALU) in Warps (for e.g. Matrix Multiplication) in a block (SRAM level, streaming multiprocessor) and blocks in the grid (multiple streaming multiprocessors) - additionally there can be looping levels where the parallelization is resolved to a simple loop. Dimensions are: $b$: batch, $t$: time, $g$: gates, $s/s'$: previous/new state

**Require:** Recurrent matrix $\boldsymbol{R}_{gs}$, inputs $\boldsymbol{x}_{tbg}$, biases $\boldsymbol{b}_g$
**Require:** Initial states $\boldsymbol{s}_{0bs}$
  Load $\boldsymbol{R}_{gs}, \boldsymbol{b}_g$ to registers and SRAM
  **for** $l_b$ in $L_B$ **do**
     Load $\boldsymbol{s}_{0bs}$ to registers
     **for** $t \in 0..T-1$ **do**
        **for** Matrix Tiles in Registers **do**
           Calculate and Accumulate Matrix product $\boldsymbol{y}_{tbg} = \boldsymbol{R}_{gs} \, \boldsymbol{s}_{tbs}^{(0)}$ along $s$
        **end for**
        **for** Matrix Tiles in SRAM **do**
           Load Matrix Tile of $\boldsymbol{R}_{gs}$
           Calculate and Accumulate Matrix product $\boldsymbol{y}_{tbg} = \boldsymbol{R}_{gs} \, \boldsymbol{s}_{tbs}^{(0)}$ along $s$
        **end for**
        Accumulate MatMul results $\boldsymbol{y}_{tbg}$ along $s$ in shared memory (Write, Load and Sum)
        **if** state dimension too big for SRAM **then**
           Accumulate MatMul results $\boldsymbol{y}_{tbg}$ along $s$ in HBM (Write, Grid Sync, Load, Sum)
        **end if**
        Sum Gate inputs $\boldsymbol{x}_{tbg}$ with $\boldsymbol{y}_{tbg}$ and biases $\boldsymbol{b}_g$ to gates $\boldsymbol{g}_{tbg}$
        Compute Point-wise Function $\boldsymbol{s}_{t+1bs'} = \mathcal{P}(\boldsymbol{s}_{tbs'}, \boldsymbol{g}_{tbg})$ with aligned states $s'$ and gates $g$
        Write out gates for backward pass and new states to HBM
        Grid-Level Sync (for new states to be available across the whole grid)
     **end for**
  **end for**

---

## 5.3 TRITON IMPLEMENTATION

With FlashRNN we also implement a Triton[4] variant of the fused FlashRNN kernel. Triton is a domain specific language and compiler for parallel programming that provides a Python-based environment for writing custom GPU kernels.

For the Triton kernel we parallelize the computation over two dimensions the batch dimension and the head dimension. See Appendix E for a detailed description of the Triton implementation in Algorithm 5 of the FlashRNN algorithm. As described in section 4.2 we partition the embedding dimension into multiple heads and compute each head in parallel in different programs (or thread blocks) with no synchronization in between these programs. In Triton, each program (which corresponds to a thread block in CUDA) will hold its recurrent weight matrix $\boldsymbol{R}_{head}$ and bias $\boldsymbol{b}_{head}$ in SRAM. In contrast to CUDA, Triton gives no access to registers on the GPU. Therefore, we cannot apply the custom caching strategy of the fused CUDA kernels and instead rely on Triton for managing the shared memory and register cache. Additionally, there is no (grid) synchronization between programs in Triton, which makes it impossible to communicate values between different programs over HBM. In section 6.1, we find that this poses a limitation on the maximum head dimension of 128 for the forward pass and 64 for the backward pass on a NVIDIA H100 GPU.

The recurrent matrix multiplication in equation 1 and 3 is implemented with Triton's matrix multiply operation `tl.dot`, which gives an interface to the Tensor Core units on GPUs. In Triton, the minimum block size of these matrix multiplies is 16x16, which gives a limit on the minimum batch size. In practice, we enable smaller batch sizes by padding zeros at the cost of efficiency.

---

[4]https://triton-lang.org

### 5.4 AUTOMATIC TUNING OF TILING AND LOOPING DIMENSIONS

While Algorithm 1 describes the algorithmic behaviour, the tile, block and grid sizes and loop iterations depend on the specific hardware architecture, i.e. the number of computational super-units (streaming multiprocessors), the SRAM per super-unit, the sizes of matrix-multiplication units, threads (warps and threads) per super-unit and the number of registers per thread. On NVIDIA H100s (and most other NVIDIA GPUs), there is a varying amount of registers per thread, depending on the block size used, while the total number of registers per SM is physically fixed.

These physical constraints can now be reformulated as equalities, inequalities and divisibility constraints inside an integer constraint satisfaction problem (integer CSP). Typically this optimization is done via polyhedral constraint optimization in compilers (Baghdadi et al., 2018). For solving these constraints in FlashRNN, we implement an efficient solver ConstrINT in Python for general integer CSPs going over large number ranges and including the notion of divisibility constraints, which are needed to model the minimal matrix sizes.

For more details on the solution algorithm, see Appendix Section C.

## 6 EXPERIMENTS

In Section 6.1, we benchmark the runtime of our FlashRNN kernels and compare against the LSTM and Attention implementations provided in PyTorch. In Section 6.2, we measure training time with FlashRNN kernels on language modeling. Finally, in Section 6.3 we confirm that traditional RNNs like LSTM and sLSTM implemented in FlashRNN can solve state tracking problems.

### 6.1 RUNTIME BENCHMARK

We evaluate the runtime of all backends of our FlashRNN library that implement an LSTM:

- **CUDA fused:** CUDA implementation that fuses matrix multiplication and pointwise operations of the LSTM in a single kernel that is persistent over all time iterations.
- **CUDA alternating:** CUDA implementation that implements the time loop in C++ and alternates between a matrix multiply kernel and a LSTM pointwise kernel.
- **Triton fused:** Triton implementation that fuses matrix multiplication and pointwise operations similar to CUDA fused.
- **Vanilla PyTorch:** PyTorch implementation of the LSTM operation with our custom backward pass implementation, which is faster than the PyTorch autograd backward pass. We do not use `torch.compile` due to very long compile times.

We compare our backends to two references from PyTorch and the haste library (Sharvil, 2020):

- **FlashAttention2:** PyTorch Attention[5] with FlashAttention2 backend. Note that FlashAttention2 is not a recurrent operation and can be parallelized across batch, head, and sequence dimension on the GPU. FlashAttention2 does not fall into the category of RNNs, which FlashRNN aims to speed up, and is not able to solve state tracking tasks. Therefore, in our benchmarks it should be seen as a widely adopted reference to better interpret the runtimes instead of a direct baseline that we aim to outperform.
- **nn.LSTM:** PyTorch LSTM with NVIDIA cuDNN as backend. In contrast to our FlashRNN LSTM, `nn.LSTM` also integrates the gate pre-activation computation into the function call (not kernel call), which we do not (see Section 3). In Section H.4 in the appendix, we provide a comparison to the combination of a linear layer and our FlashRNN LSTM kernel with `nn.LSTM`. Moreover, `nn.LSTM` does not support multiple heads on the embedding dimension as described in Section 4.2. `nn.LSTM` always uses a single head.
- **haste:** The `haste` library is an implementation of LSTM and GRU and variations in CUDA, using alternating kernels between pointwise and matrix multiplication operations.

---

[5]`https://pytorch.org/docs/stable/generated/torch.nn.functional.scaled_dot_product_attention.html`

Its last release was in 2020, with no compilation support for Ampere or later architectures in the standard setting[6]. It solely supports float32 and float64 precision and does not have a multi-head option.

**Setup.** We assess the impact of the input dimensions batch size (B), sequence length (T) and head dimension (DH) and number of heads (NH). The number of heads together with the head dimension give the embedding dimension $d = $ NH $\times$ DH. Except for PyTorch `nn.LSTM` we run all runtime experiments with bfloat16 precision. For `nn.LSTM` we use float16 precision, since this precision yielded the fastest runtimes. For every runtime measurement we do 25 warmup iterations and then report the average across 1000 iterations on NVIDIA H100 GPUs. We use PyTorch 2.4 and with CUDA version 12.4 for our experiments. Further details and additional experiments can be found in Section H in the appendix.

**Head dimension.** We measure the runtime of all of our FlashRNN kernels and our two references FlashAttention2 and PyTorch `nn.LSTM` for different head dimensions. We fix the embedding dimension $d = $ NH $\times$ DH to 768 and vary the head dimension from 16 to 768. We use batch size 16 and sequence length 1024. In Figure 2, we report the runtime of each the forward pass only on the left and the forward combined with the backward pass. FlashAttention2 does not allow for head dimension larger than 256, due shared memory limitation. The PyTorch `nn.LSTM` does not support multiple heads or blockdiagonal recurrent matrices. Therefore, we only report the runtime for a single head of dimension 768, including the gate pre-activation computation. At this dimension, `nn.LSTM` is about 3 times faster than CUDA fused. The Triton kernels are limited to head dimension 128 and 64, but are about two times faster than CUDA fused for small head dimensions 16 and 32. The fused CUDA kernels support all head dimensions up to 768 (actually more, see Appendix Section H.1) and are about two to three times faster than the alternating kernels.

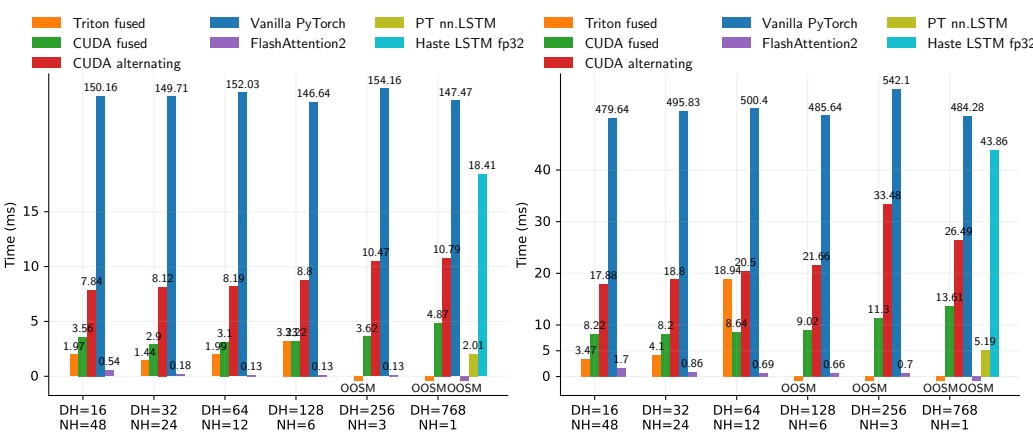

Figure 2: LSTM Runtime for different head dimensions (DH) and number of heads (NH) on a NVIDIA H100. Overall embedding dimension is fixed at 768. We use batch size 16 and sequence length 1024. **Left:** Forward pass. **Right:** Forward + backward pass.

**Batch size.** We measure the runtime of all LSTM kernels while varying the batch size (B) from 2 to 256 at sequence length 1024. Figure 3 shows the results for NH=12 heads with head dimension DH=64. The CUDA fused backend is optimized for smaller batch sizes and shows a 2x speed up over the alternating backend for batch sizes up to 32. For larger batch sizes than 128 CUDA alternating is faster. Figure 4 shows the results for a single head with head dimension DH=768. At this head dimension CUDA fused is still faster than CUDA alternating up to batch size 32. For larger batch sizes, CUDA alternating is more than two times faster. Comparing to the PyTorch `nn.LSTM`, we find for medium batch sizes from 8 to 64 it is about 2-3 times faster than and CUDA fused and for larger batch sizes about about 30% faster than CUDA alternating.

---

[6] https://github.com/lmnt-com/haste

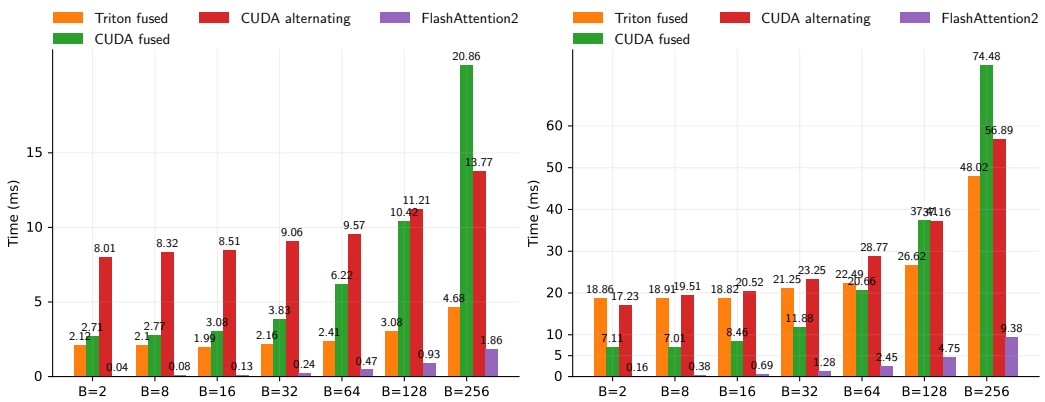

Figure 3: LSTM Runtime for different batch sizes (B) on a NVIDIA H100. We use 12 heads with head dimension 64 and sequence length 1024. **Left:** Forward pass. **Right:** Forward + backward pass.

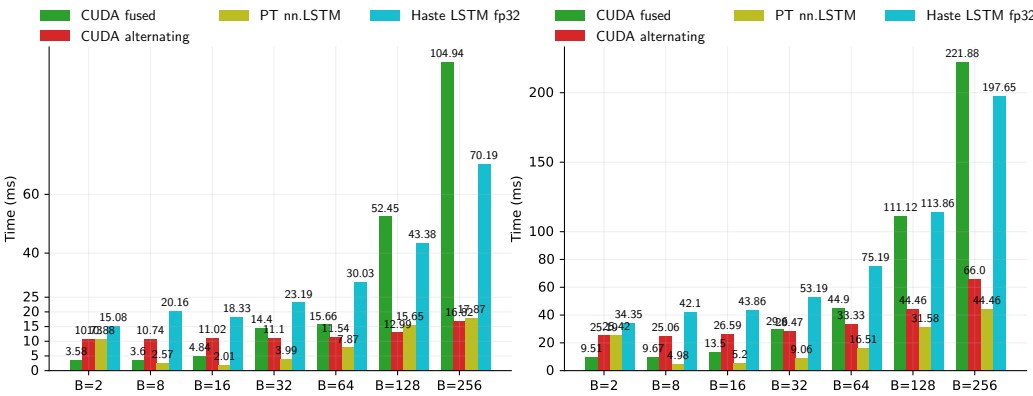

Figure 4: LSTM Runtime for different batch sizes (B) on a NVIDIA H100. We use one head ($d_{head} = 768$) and sequence length 1024. **Left:** Forward pass. **Right:** Forward + backward pass.

**Additional Runtime Experiments.** In section H.3 in the appendix, we include experiments on varying sequence lengths. We see the expected linear runtime scaling for our FlashRNN kernels and validate that the above findings transfer to longer sequences. In addition, in section H.4 we compare the FlashRNN LSTM kernel in combination with a linear layer that computes the gate pre-activations externally to the PyTorch `nn.LSTM` baseline, which integrates the gate pre-activation computation. We find that the gate pre-activation computation has only marginal impact on the overall runtime. Finally, in section H.5, we provide all runtime results also for the sLSTM (Beck et al., 2024).

## 6.2 LANGUAGE MODELING

Even though we do no expect traditional RNNs to outperform Transformers, the language modeling setting serves as an important benchmark for speed on larger scales. Here, we train models at the 165M parameter scale for a Llama-style Transformer without weight tying, i.e. 12 Transformer blocks with Pre-LayerNorm and a Swish-Gated MLP after the attention layer. We replace attention with FlashRNN LSTM and sLSTM layers for a speed comparison. The results show a slowdown of roughly 25 % over attention for equal head dimensions or 140 % for one RNN head, see Table 6.2 (H100) and Appendix Table I (A100). In our experiments, we also compare to the cuDNN implementation of LSTM integrated in PyTorch (torch.nn.LSTM). While it's integration into PyTorch is considerably faster, there are numerical differences to the FlashRNN implementation. With same initialization, FlashRNN LSTMs converge faster in our language experiments (both bfloat16 and

| Model | Heads | Param. (M) | Train Time (h) | Median Step (s) | Val PPL (val) |
|---|---|---|---|---|---|
| LSTM CUDA fused | 1 | 190 | 9.9 | 0.535 | 22.1 |
| LSTM CUDA altern. | 1 | 190 | 10.8 | 0.575 | 21.9 |
| LSTM PT nn.LSTM | 1 | 190 | 4.5 | 0.285 | 25.8 |
| LSTM CUDA fused | 12 | 164 | 5.9 | 0.325 | 22.2 |
| LSTM CUDA altern. | 12 | 164 | 9.6 | 0.511 | 22.1 |
| sLSTM CUDA fused | 1 | 190 | 10.1 | 0.543 | 21.3 |
| sLSTM CUDA altern. | 1 | 190 | 10.9 | 0.577 | 21.4 |
| sLSTM CUDA fused | 12 | 164 | 6.8 | 0.342 | 21.7 |
| sLSTM CUDA altern. | 12 | 164 | 9.7 | 0.509 | 21.8 |
| Transformer | 12 | 162 | 2.9 | 0.190 | 17.9 |

Table 1: 165M Model training on 15B tokens of SlimPajama on 8xH100s with two gradient accumulation steps.

float32), even though the differences in a single kernel call are at the expected levels of numerical precision. This deviation should be investigated further and suggests the use of FlashRNN even for the established LSTM architecture. We provide an analysis of our kernel precision compared to a float64 baseline in section H.6.

For larger models, we expect local batch sizes to be smaller and the effective speed difference for fused kernels to be higher compared to the alternating version - as measured in Section 6.1.

### 6.3 STATE TRACKING TASK

To show state tracking capabilities of traditional RNNs in contrast to Transformers and State Space Models experimentally, we train our implementation on the Parity task and evaluate on longer sequences to measure extrapolation capabilities (Zhou et al., 2024). This serves as a litmus test for state tracking capabilities (Merrill et al., 2024).

| Model | Transformer | Mamba | mLSTM | Elman | GRU | LSTM | sLSTM |
|---|---|---|---|---|---|---|---|
| Acc (Ext.) | 0.52 | 0.56 | 0.54 | 1.00 | 1.00 | 1.00 | 1.00 |

Table 2: Parity Task in Sequence Extrapolation: Transformers, State Space Models and mLSTM fails at this task (close to random chance at 0.5), while traditional recurrent models can learn to extrapolate. Extrapolation accuracies are averaged over three seeds for the best learning rate.

## 7 CONCLUSION

The FlashRNN library serves as a fast and extendable implementation of traditional RNNs with a recurrent connection or memory mixing. It extends RNNs with the multi-head paradigm introduced by Beck et al. (2024) for sLSTM. FlashRNN provides a speed-up of up to 50x over vanilla PyTorch implementations of RNNs and may serve as a backbone for future RNN architectures that have a recurrent connection.

FlashRNN implements two variants, an alternating version switching between point-wise and matrix-multiplication kernels and a fused implementation - optimizing memory transfers, while using hardware-optimized matrix-multiplication. The second leads to a further 3-4x speed-up over the alternating option for small batch sizes. The implementation auto-optimizes its internal sizes for different cache levels via the ConstrINT library - a custom library solving general integer constraint satisfaction problems with equality, inequality and divisibility constraints. This library may be re-used for other optimization problems regarding cache sizes on hardware platforms and beyond.

We show that with FlashRNN, traditional RNNs are not too far in speed from Transformers in practice, even though they are not parallelizable along the sequence dimension. In the future, it may be optimized to leverage asynchronous memory operations and inter-SRAM connections - recent hardware features that promise further speed ups not realized in this work.

ACKNOWLEDGMENTS

We thank Markus Spanring, Maxim Milakov, Pieter-Jan Hoedt, Günter Klambauer and Fabian Paischer for helpful discussions and feedback.

We acknowledge EuroHPC Joint Undertaking for awarding us access to Karolina at IT4Innovations, Czech Republic, MeluXina at LuxProvide, Luxembourg, Leonardo at CINECA, Italy and LUMI at CSC, Finland. The ELLIS Unit Linz, the LIT AI Lab, the Institute for Machine Learning, are supported by the Federal State Upper Austria. This research was funded in whole or in part by the Austrian Science Fund (FWF) [10.55776/COE12]. We thank the projects INCONTROL-RL (FFG-881064), PRIMAL (FFG-873979), S3AI (FFG-872172), DL for GranularFlow (FFG-871302), EPILEPSIA (FFG-892171), FWF AIRI FG 9-N (10.55776/FG9), AI4GreenHeatingGrids (FFG-899943), INTEGRATE (FFG-892418), ELISE (H2020-ICT-2019-3 ID: 951847), Stars4Waters (HORIZON-CL6-2021-CLIMATE-01-01). We thank NXAI GmbH, Audi.JKU Deep Learning Center, TGW LOGISTICS GROUP GMBH, Silicon Austria Labs (SAL), FILL Gesellschaft mbH, Anyline GmbH, Google, ZF Friedrichshafen AG, Robert Bosch GmbH, UCB Biopharma SRL, Merck Healthcare KGaA, Verbund AG, GLS (Univ. Waterloo), Software Competence Center Hagenberg GmbH, Borealis AG, TÜV Austria, Frauscher Sensonic, TRUMPF and the NVIDIA Corporation.

ETHICS STATEMENT

We use an open dataset (SlimPajama) that uses publicly crawled internet data for Language Model training. Our implementation speeds up a certain class of Machine Learning models. This may reduce the environmental impact of the research field, in case these architectures remain important in future research. Also, it may speed up development of Machine Learning research in the direction of recurrent sequence modeling with state tracking capabilities. The further implications of these impacts may or may not be a benefit for society.

REPRODUCIBILITY STATEMENT

We provide the source code for your implementations along with this paper. The detailed training setup for speed tests is described in Section 6.1. For Language Modeling this setup description is provided in Appendix Section J and uses the open SlimPajama dataset, for the parity task experiments in Appendix Section K, the training and test data can be synthetically generated using the mentioned distributions.
The observed deviations in language model training compared to the PyTorch LSTM based on cuDNN should be further investigated. The results on A100 and H100, as well as across our different kernels are within the expected small-scale numerical deviations.
The code is released here: `https://github.com/NX-AI/flashrnn`, with the ConstrINT library in *flashrnn/autotune/constrint.py* as single-file implementation and optional caching.

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

## A RNN VARIANTS WITH MEMORY MIXING / RECURRENT CONNECTIONS MODELED IN FLASHRNN

**Elman RNNs**  (Elman, 1990) Number of states: 1, Number of gates: 1

$$s_t^{(0)} = \tanh\left(g_t^{(0)}\right) \tag{8}$$

Here, we omit as well a possible post-processing of the sequence that does not inter-mix states of different time steps. This can as well be parallelized for offline training.

**LSTM**  (Hochreiter & Schmidhuber, 1997; Gers et al., 1999) Number of states: 2, Number of gates: 4

States: $h_t = s_t^{(0)}$ hidden state, $c_t = s_t^{(1)}$ cell state

Gates: $z_t = g_t^{(0)}$ cell input, $f_t = g_t^{(1)}$ forget gate, $i_t = g_t^{(2)}$ input gate, $o_t = g_t^{(3)}$ output gate

$$h_t = \sigma\left(o_t\right)\tanh\left(c_t\right) \tag{9}$$
$$c_t = \sigma\left(f_t\right)c_{t-1} + \sigma\left(i_t\right)\tanh\left(z_t\right) \tag{10}$$

**GRU**  (Cho et al., 2014) Number of states: 1, Number of gates: 4 (in the definition of this paper)

States: $s_t^{(0)}$ hidden state

Gates: $z_t = g_t^{(0)}$ cell input, $r_t = g_t^{(1)}$ forget gate, $n_t = g_t^{(2)}$ input gate, $o_t = g_t^{(3)}$ output gate Here, the $n_t$ gate is not dependent on the previous state, whereas the $g_t$ gate is not dependent on the input. This behavior can be modeled in FlashRNN as well.

$$h_t = \sigma\left(z_t\right)h_{t-1} + \left(1 - \sigma\left(z_t\right)\right)\tanh\left(n_t + \sigma\left(r_t\right)\tanh(g_t)\right) \tag{11}$$

**sLSTM**  (Beck et al., 2024) Number of states: 4, Number of gates: 4 States: $h_t = s_t^{(0)}$ hidden state, $c_t = s_t^{(1)}$ cell state, $n_t = s_t^{(2)}$ normalizer state, $m_t = s_t^{(3)}$ stabilizer state

Gates: $z_t = g_t^{(0)}$ cell input, $f_t g_t^{(1)}$ forget gate, $i_t = g_t^{(2)}$ input gate, $o_t = g_t^{(3)}$ output gate

$$h_t = \sigma\left(o_t\right)\frac{c_t}{n_t} \tag{12}$$
$$c_t = \exp((\log\sigma\left(f_t\right) + m_{t-1} - m_t)c_{t-1} + \exp\left(i_t - m_t\right)\tanh\left(z_t\right) \tag{13}$$
$$n_t = \exp((\log\sigma\left(f_t\right) + m_{t-1} - m_t)n_{t-1} + \exp\left(i_t - m_t\right) \tag{14}$$
$$m_t = \max\left(\log\sigma\left(f_t\right) + m_{t-1}, i_t\right) \tag{15}$$

## B FLASHRNN ALGORITHM IN DETAIL

In the following algorithm, $\mathbb{R}^{a,b\times(c\cdot d)}$ means this is seen as a matrix tile of size $b \times (c \cdot d)$, where $a$ is an additional outer index (typically time $t$ or states $s$), which denotes that this is used as a separate outer dimension. The merged dimension $(c \cdot d)$ is shown merged as it is used in matrix multiplications, but is split (typically into gates) in the pointwise function. The dimensions are: $t$: time, $s$: states, $g$: gates, $b$: batch dimension, $d$: head dimension (abbreviated from $d_{head}$).

For accumulation of the recurrent matrix product there are two matrix dimensions: The dimension along the previous state $\tilde{s}$ (of total size $d$) and the dimension along the new gates $\tilde{g}$ of total size $g \cdot d$, since for every of the $d$ RNN cells there are $g$ gates. Multiple heads are parallelized either sequentially or over multiple blocks in the grid $B_H$, but we omit this for clarity here.

Tiling along one axis $A$ happens as an elementary tile size within a warp $E_A$, multiple warps $W_A$ in a thread block, multiple blocks $B_A$ within the grid, and sequentially via a loop $L_A$. The total size has to satisfy $S_A = E_A \times W_A \times B_A \times L_A$. The typical elementary size $E_A$ usable for matrix multiplications in bfloat16 is $E_A \in \{8, 16, 32\}$ for an outer dimension and $E_A = 16$ for the accumulating dimension. The number of elementary tiles is $T_A = \frac{S_A}{E_A} = W_A \times B_A \times L_A$. To optimize speed internally, we use memory padding to minimize memory bank conflicts and coalesced memory loading.

---

**Algorithm 2** FlashRNN-fused forward pass

---

Tiling across blocks should be kept minimal along accumulating dimensions, and can be extended along parallelizing dimensions (here gate dimension). So ideally $B_S \ll B_G$. Indices $\tilde{b}, \tilde{s}, \tilde{g}$ are implicitly updated from loop indices $l_B, l_S, l_G$ incorporating the respective warp/block indices.

**Require:** Recurrent matrix $\boldsymbol{R}^\top \in \mathbb{R}^{d \times (d \cdot g)}$, inputs $\boldsymbol{x} \in \mathbb{R}^{t, b \times (d \cdot g)}$, biases $\boldsymbol{b} \in \mathbb{R}^{d \cdot g}$

**Require:** Initial state $\boldsymbol{s}^{(0)} \in \mathbb{R}^{s, 1, b \times d}$

**Require:** Tiling dimensions for the grid $[B_G, B_S, B_B * B_H]$ and block size $[32 \times W_G, W_S, W_B]$

    Divide $\boldsymbol{R}^\top$ into $[T_S, T_G] = [L_S \times W_S \times B_S, L_G \times W_G \times B_G]$ tiles $\boldsymbol{R}_{\tilde{s}, \tilde{g}}^\top \in \mathbb{R}^{16 \times (16 \text{ or } 32)}$ with $\tilde{s} \in \{1..T_S\}, \tilde{g} \in \{1..T_G\}$ along the state (first) and gate (second) dimension

    Divide the bias $\boldsymbol{b}$ into $[L_G \times W_G]$ tiles along the gate dimension as $\boldsymbol{b}_{\tilde{g}} \in \mathbb{R}^{(16 \text{ or } 32)}$

    Load tiles $\boldsymbol{R}_{\tilde{s}, \tilde{g}}^\top, \boldsymbol{b}_{\tilde{g}}$ from HBM into registers ($[L_S, L_G] / [L_G]$ per warp) and potentially SRAM across multiple thread blocks $[B_S, B_G, B_B]$

    **for** $l_B$ in $\{1..L_B\}$ **do**

        Load from initial state $\boldsymbol{s}^{(0)}$ a batch tile $\boldsymbol{s}_{\tilde{b}\tilde{s}}^{(0)}$

        **for** $t \in 0..T-1$ **do**

            **for** $l_G \in 1..L_G$ **do**

                Initialize MatMul result $\boldsymbol{y} \in \mathbb{R}^{E_B \times E_G}$ with zero in registers.

                **for** $l_S \in 1..L_S^{(\text{reg})}$ **do**

                    Load state matrix tile $\boldsymbol{s}_{0t\tilde{b}\tilde{s}}$ from HBM

                    Calculate and Accumulate Matrix Product $\boldsymbol{y} = \boldsymbol{y} + \boldsymbol{s}_{0t\tilde{b}\tilde{s}} \boldsymbol{R}_{\tilde{s}\tilde{g}}^\top$ along $\tilde{s}$

                **end for**

                **if** $L_S^{(\text{reg})} \leq L_S$ **then**

                    **for** $l_S \in 1..L_S^{(\text{SRAM})}$ **do**

                        Load state matrix tile $\boldsymbol{s}_{0t\tilde{b}\tilde{s}}$ from HBM

                        Load recurrent matrix tile $\boldsymbol{R}_{\tilde{s}, \tilde{g}}^\top$ from SRAM

                        Calculate and Accumulate Matrix Product $\boldsymbol{y} = \boldsymbol{y} + \boldsymbol{s}_{0t\tilde{b}\tilde{s}} \boldsymbol{R}_{\tilde{s}\tilde{g}}^\top$ along $\tilde{s}$.

                  **end for**

                **end if**

                Store MatMul result $\boldsymbol{y}$ in SRAM

                Block Level Sync

                **for** $w_S$ in $1..W_S - 1$ **do**

                    Load other MatMul result $\tilde{\boldsymbol{y}}_{\tilde{s}}$

                    Accumulate MatMul result $\boldsymbol{y} = \boldsymbol{y} + \tilde{\boldsymbol{y}}_{\tilde{s}}$

                **end for**

                Block Level Sync

                Store MatMul result $\boldsymbol{y}$ in SRAM

                **if** $B_S \geq 1$ **then**

                  # Reorder tiling here for coalescing memory access and optimal work partitioning

                  Store MatMul result $\boldsymbol{y}$ in HBM

                  Grid Level Sync

                **end if**

                # Reorder tiling here with in a block for one thread per point-wise op.

                Load Gate inputs $\boldsymbol{x}_{t\tilde{b}\tilde{g}}$ from HBM

                Load MatMul result $\boldsymbol{y}$ from SRAM

                Add $\boldsymbol{g} = \boldsymbol{x}_{t\tilde{b}\tilde{g}} + \boldsymbol{b}_{\tilde{g}} + \boldsymbol{y}$

                **for** $b_s \in 2..B_S$ **do**

                    Load other MatMul result $\tilde{\boldsymbol{y}}_{\tilde{s}}$ from HBM

                    Add $\boldsymbol{g} = \boldsymbol{g} + \tilde{\boldsymbol{y}}_{\tilde{s}}$

                **end for**

                Point-wise Update $\boldsymbol{s}_{t+1\tilde{b}\tilde{s}'} = \mathcal{P}(\boldsymbol{s}_{t\tilde{b}\tilde{s}'}, \boldsymbol{g}_{t\tilde{b}\tilde{g}})$ with aligned states $\tilde{s}'$ and gates $\tilde{g}$

                Write out gates $\boldsymbol{g}_{t\tilde{b}\tilde{g}}$ to HBM for backward pass

                Write out new states $\boldsymbol{s}_{t+1\tilde{b}\tilde{s}'}$ to HBM

            **end for**

            Grid-Level Sync (for new states to be available across the whole grid)

        **end for**

    **end for**

---

## C    ConstrINT resolution algorithms

To model the hardware constraints, we define IntegerVariables, e.g. a variable describing a tiling size in the FlashRNN algorithm or a constant that defines the total SRAM for one streaming multiprocessor. These can attain a set of numbers (domain), e.g. initially a large range for a so far unconstrained tiling size or a certain value for a constant. These variables can be composed to terms via addition and multiplication, and these terms can be constrained via equalities, inequalities and divisibility constraints.

---

**Algorithm 3** ConstrINT Resolution

---

**Require:** Input Constants / Variables
**Require:** Resolution Variables with Heuristic
**Require:** Equality, Inequality and Divisibility Constraints
  Generate intermediate/background variables for terms that propagate constraints
  Reach arc consistency via "Global ARC-Reduce"
  **if** any variable has empty domain $|D_V| = 0$ **then**
    **return** "No Solution viable"
  **end if**
  Sort values for each Resolution Variable via Heuristic
  **while** any Resolution Variable has domain $|D_V| > 1$ (not fixed) **do**
    Choose Variable via Heuristic, Increase Index Count
    **if** Lowest Order Variable has empty domain **then**
      **return** "No Solution viable"
    **end if**
    Set Variable Value via Heuristic
    Reach arc consistency via Global ARC-Reduce
    **if** any variable has empty domain $|D_V| = 0$ **then**
      Backtrack
    **end if**
  **end while**
  **return** Solution

---

**Algorithm 4** ConstrINT Global ARC-Reduce

---

**Require:** Expression Parse Tree of Constraints and Variables
  Status=Not Converged
  **while** Change in Root IntegerVariable or NotConverged **do**
    Propagate Restrictions to SubTerms of Expression
    **for** Sub-Term in Root-Expression **do**        ▷ Top-Down Application of Constraints
      Apply Global ARC-Reduce on Sub-Term - Get changes
    **end for**
    **if** any change in values for Sub-Term **then**      ▷ Bottom-Up Application of Constraints
      Propagate Restriction from Sub-Terms to Root Variable
      Status=Not Converged
    **else**
      Status=Converged
    **end if**
  **end while**
  **return** change in Root IntegerVariable

---

Specific resolution variables additionally have a heuristic added that defines the behaviour of iteration for choosing among possible values. If the domain of all resolution variables is reduced to a single number, this is a solution. The heuristic gives an order of these variables and for each variable, if smaller or larger values are expected to result in a "better" solution. This helps optimization as there might be many possible solutions, but certain ones promise most speed-ups (e.g. using most TensorCores).

At the lowest level, a term is composed of two IntegerVariables (or intermediate variables), so constraints on it propagate down to the two summand or factor IntegerVariables. Equality, Inequality

and Divisibility constraints propagate to the contained terms as well. For example, since all numbers are strictly positive the upper bound on a sum of two IntegerVariables applies to both the summands - minus one. Applying the constraints iteratively upwards and downwards in the expression parse tree until convergence (i.e. no change for any variable) leads to an arc-consistent state, which we call "Global ARC-Reduce". The binary "ARC-Reduce" algorithm is part of the "AC-3", a constraint satisfaction problem solver for a more general setting (Mackworth, 1977). An arc-consistent state might still have no solution, it is merely a super-set of all possible solutions. Based on the heuristic the ConstrINT algorithm applies a depth-first tree search with "Global ARC-Reduce" application at each step and backtracking for an empty solution domain.

## D CONSTRINT KERNEL OPTIMIZATION

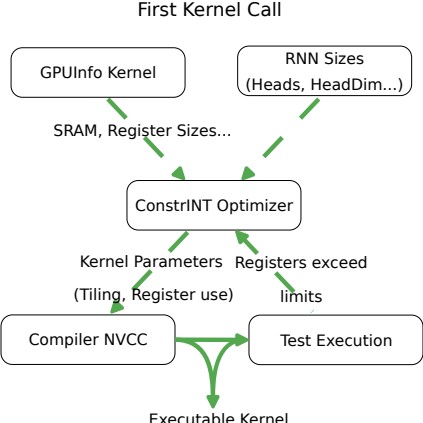

Figure 5: JIT Optimization procedure for first kernel call. RNN parameters and GPU hardware info are processed by ConstrINT for a feasible kernel parametrization. Since register use cannot fully be predicted in advance, register use is iteratively optimized with feedback from the compiler. Subsequently, the kernel is cached as well as the intermediate optimization solutions.

The fused kernel described in Section 5.2 and in more detail in Appendix Section B has certain external parameters, which have to be set correctly for the kernels to be runnable and fast. The main constraints here are the size limitations of registers and SRAM. For a large hidden or head dimension $d$, e.g. $d = 768$, the recurrent weight matrix for an LSTM has the size $4 \times 768 \times 768 \times 2\,\text{B} \approx 4\,\text{MB}$. However, for an H100 GPU, the register file size is $256KB$ and the SRAM / shared memory is up to $228KB$ per SM / block. Therefore, this matrix needs to be sharded over multiple SMs in a cooperative grid group that can synchronize on the grid level. In particular, the different variables defined in Appendix Section B: $E_B, W_B, B_B, L_B, E_G, W_G, B_G, L_G, E_S, W_S, B_S, L_S^{(\text{reg})}, L_S^{(\text{SRAM})}$ imply certain sizes of registers and SRAM via a polynomial function. ConstrINT optimizes these variables to fit within the boundaries of the hardware and to achieve a reasonable speed, by ordering the variables and applying a heuristic on their values. For example, the gate dimension $\tilde{g}$ in the forward pass is a pure parallelization, whereas the state dimension $\tilde{s}$ is accumulated over. Accumulation necessitates additional memory operations and synchronization that make the execution slower, which is not the case for the purely parallel dimension. Therefore the $B_G$ variable is maximized, while $B_S$ is minimized during the constraint satisfaction solution search.

Furthermore, it is a priori not clear how many registers can be used by the kernel for storing additional variables. Therefore, ConstrINT is used in a feedback loop together with the compiler performing a binary search for the largest attainable register size.

Given a more detailed understanding of the kernel as shown in Section B, ConstrINT variables can be fixed to different values for a manual optimization of a specific kernel size. ConstrINT will automatically optimize all other variables using the hardware limits and given heuristics.

An example, where this is used in the standard version is the block size (threads / warps per thread block). While the maximum could be 1024 on NVIDIA GPUs, we set this manually to a fourth. This is usually faster, while not restricting the usable memory.

# E DETAILS ON TRITON IMPLEMENTATION

Algorithm 5 provides details on the Triton implementation. It shows the computation for a single program or thread block, which computes one head of dimension $d$ and block $B_b$ of the batch dimension $b$. We run a grid of $(n_{head} \times \frac{b}{B_b})$ of these programs in parallel for FlashRNN forward pass in Triton. We load the recurrent weights $\boldsymbol{R}$ and biases $\boldsymbol{b}$ only once from HBM to SRAM and keep them in SRAM throughout the time loop.

On a higher level the main differences to the CUDA implementation in algorithm 1 are that in CUDA we can use multiple thread blocks for a single head and we can force the kernel to keep the recurrent weights $\boldsymbol{R}$ in registers instead of SRAM. One can see this difference for example in the kernel launch grid, which parallelizes only over number of heads and blocks of batch size in Triton, while it has two more parallelization dimensions in CUDA (see Algorithm 1).

---

**Algorithm 5** Triton FlashRNN Forward Pass

---

**Require:** Recurrent weights $\boldsymbol{R}^{(j)} \in \mathbb{R}^{d \times d}$, biases $\boldsymbol{b}^{(j)} \in \mathbb{R}^d$ for gates $j$ and inputs $\boldsymbol{x}_t^{(j)} \in \mathbb{R}^d$ for
    gates $j$ and timesteps $t = 1..T$;
    Initial states $\boldsymbol{s}_0^k \in \mathbb{R}^d$
    Load $\boldsymbol{R}^{(j)} \in \mathbb{R}^{d \times d}$, biases $\boldsymbol{b}^{(j)} \in \mathbb{R}^d$
    Load initial states $\boldsymbol{s}_0^{(k)} \in \mathbb{R}^d$
    **for** timestep $t = 1..T$ **do**
        Load inputs $\boldsymbol{x}_t^{(j)}$
        Compute gate preactivations $\boldsymbol{g}_t^{(j)} = \boldsymbol{x}_t^{(j)} + \boldsymbol{R}^{(j)} \boldsymbol{s}_{t-1}^{(0)} + \boldsymbol{b}^{(j)}$
        Compute pointwise operations $\boldsymbol{s}_t^{(i)} = \mathcal{P}^{(i)} \left( \{\boldsymbol{s}_{t-1}^{(k)}\}_k, \{\boldsymbol{g}_t^{(j)}\}_j \right)$
        Store states $\boldsymbol{s}_t^{(i)}$
        **if** Store output gates **then**
            Store gates $\boldsymbol{g}_t^{(j)}$
        **end if**
        $\boldsymbol{s}_{t-1}^{(i)} = \boldsymbol{s}_t^{(i)}$
        $\boldsymbol{g}_{t-1}^{(j)} = \boldsymbol{g}_t^{(j)}$
    **end for**
    **return** States $\boldsymbol{s}_{0:T}^{(i)}$, gates $\boldsymbol{g}_{1:T}^{(j)}$

---

# F COMPUTATIONAL COMPLEXITY

Traditional RNNs go over the sequence step by step, while applying a recurrent matrix multiplication and a pointwise activation function at each step. For the back-propagation in time, all past state values are usually stored. In this paper, we implement the head-wise notion limiting the recurrent matrix to a block diagonal form. The computational complexity is therefore: $\mathcal{O}(T \, n_{heads} \, d_{head}^2)$, with head size $d_{head}$, $n_{heads}$ the number of heads and $T$ the sequence length. The matrix vector product at each step is the dominant computational factor (for large head sizes). For inference, the memory needed is defined by the state of the RNN and is $\mathcal{O}(n_{heads} \, d_{head})$ in size.
In contrast, Attention computes a weighted sum over past inputs at each step, with the weight defined by the softmax over scalar products between query and key vectors. This leads to a computational complexity $\mathcal{O}(T^2 \, n_{heads} \, d_{head})$. The space complexity is $\mathcal{O}(T \, n_{heads} \, d_{head})$ (Vaswani et al., 2017). In conclusion, RNNs are more compressive, while their computational complexity is higher when computing only a few steps. For training with BPTT the space complexity of RNNs matches the one of Attention, as all past states have to be stored.

# G ROOFLINE ANALYSIS

As mentioned in Section 5.1, kernel speed is fundamentally limited by two factors: computation and memory bandwidth. This is usually visualized in the Roofline-Plot, showing the position of a kernel in terms of its computation throughput and arithmetic intensity. We use NVIDIA NSight Compute,

to analyse this for the alternating and fused FlashRNN LSTM kernel compared to the nn.LSTM (cuDNN) baseline on a H100-SXM:

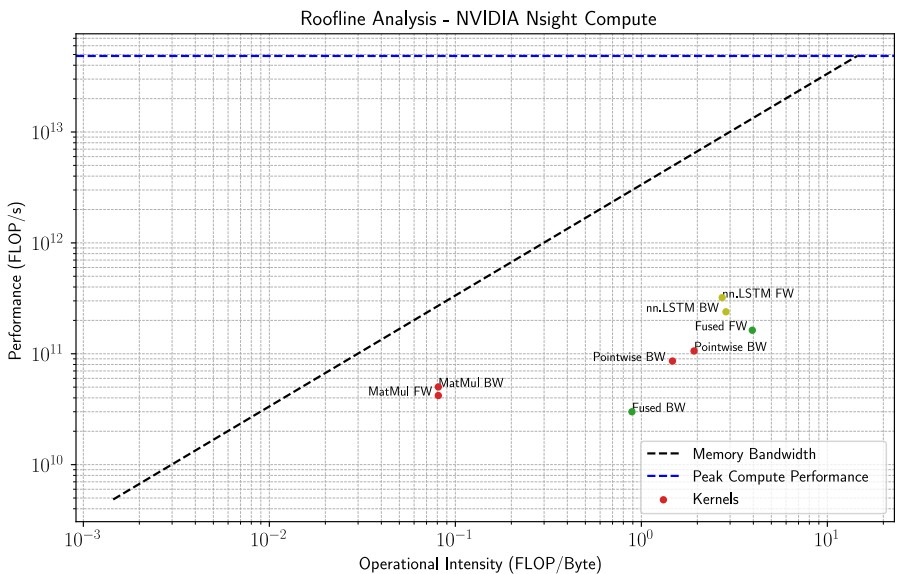

Figure 6: LSTM Kernels in the roofline plot measured with NVIDIA NSight Compute - plotting arithmetic intensity to the right, computation speed to the top. Alternating kernels show lower arithmetic intensity and performance than the fused kernels. The fused backward kernel might still be optimized compared to the nn.LSTM baseline. RNNs in general are still deep in the memory bound regime of low arithmetic intensity. The peak performance is the scalar performance limit for float32 FLOPs.

## H ADDITIONAL BENCHMARK EXPERIMENTS

### H.1 FUSED KERNEL LIMITS

Since the fused CUDA kernel of FlashRNN is based on keeping the recurrent memory matrix in registers and shared memory, there is a limit on the maximal head size - corresponding to the size of the $R$ matrix. As ConstrINT can solve these constraints automatically, there is no additional overhead other than setting this number and let it check if it works. Here, the hardware limits become visible - an RTX 3090 and A40 have 128KB of SRAM compared to 192 KB of an A100 and 228 KB of an H100 [7] [8]. For the LSTM fused kernels (forward + backward), we get the following attainable head dimensions (greater than 1280):

- RTX3090: [1280, 1312, 1344, 1440, 1536, 1600, 1632, 1728, 1824]

- A40: [1280, 1312, 1344, 1440, 1536, 1600, 1632, 1728, 1824]

- A100: [1280, 1312, 1344, 1376, 1408, 1440, 1472, 1504, 1536, 1568, 1600, 1632, 1664, 1696, 1728, 1760, 1792, 1824, 1920, 2016, 2080, 2112, 2304]

- H100: [1280, 1312, 1344, 1376, 1408, 1440, 1472, 1504, 1536, 1568, 1600, 1632, 1664, 1696, 1728, 1760, 1792, 1824, 1856, 1888, 1920, 1952, 1984, 2016, 2048, 2080, 2112, 2208, 2240, 2304, 2400, 2496, 2560, 2688]

---

[7] https://images.nvidia.com/aem-dam/en-zz/Solutions/data-center/nvidia-ampere-architecture-whitepaper.pdf

[8] https://developer.nvidia.com/blog/nvidia-hopper-architecture-in-depth/

For larger head sizes the alternating kernels can to be used, since these are not restricted in the head dimension.

## H.2 TORCH.COMPILE BASELINE

Since `torch.compile` seems to unroll the vanilla PyTorch implementation of our kernels, long sequence lengths take very long compilation times. Exemplary tests for small sequence length 64 took minutes to compile, while being only about two times faster than the vanilla PyTorch implementation without `torch.compile`. For comparison our fused kernels are up to 50 times faster.

## H.3 LSTM SEQUENCE LENGTH RUNTIME EXPERIMENTS

We confirm that the findings from Section 6.1 hold true also for varying sequence lengths from 256 to 2048. We fix the batch size to 16 and measure the runtime for 12 heads with head dimension 64 (see Figure 7) and a single head with head dimension 768 (see Figure 8). In these experiments, we see the expected linear scaling of the runtime of all LSTM kernels for increasing sequence lengths. The previous findings transfer across sequence lengths.

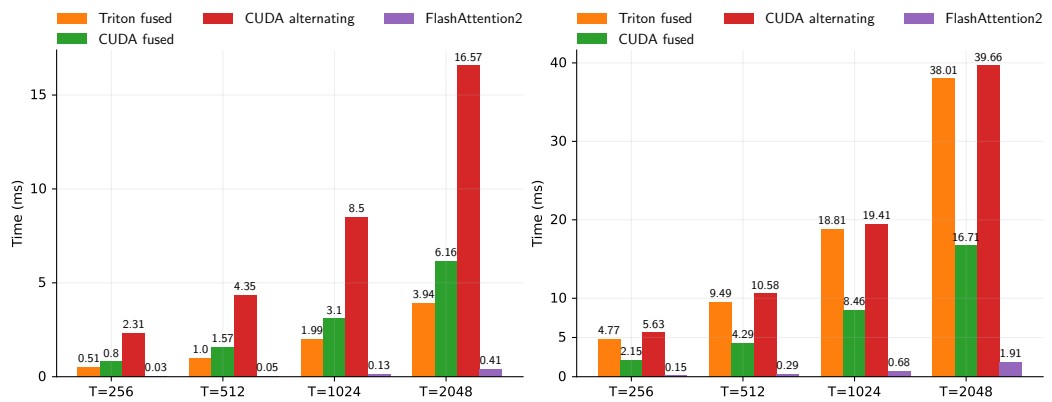

Figure 7: LSTM Runtime for different sequence lengths (T) on a NVIDIA H100. We use 12 heads with head dimension 64 and batch size 16. **Left:** Forward pass. **Right:** Forward + backward pass.

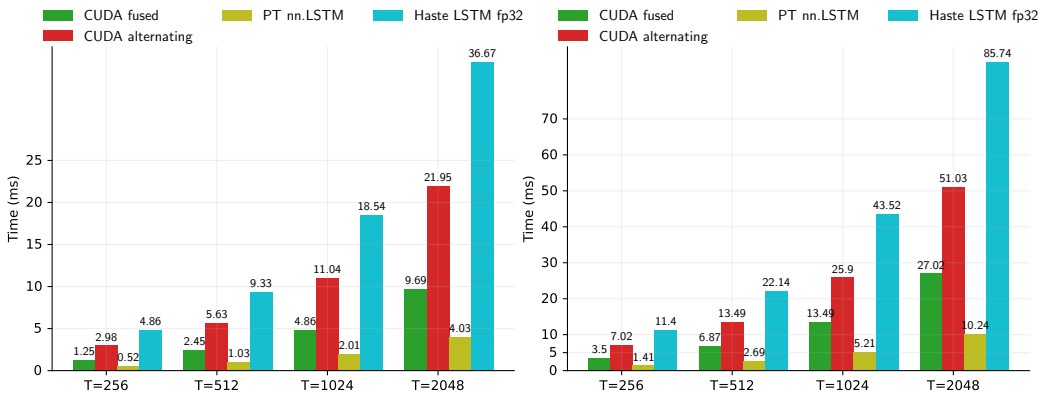

Figure 8: LSTM Runtime for different sequence lengths (T) on a NVIDIA H100. We use one head with head dimension 768 and batch size 16. **Left:** Forward pass. **Right:** Forward + backward pass.

## H.4 FlashRNN with External Gate Pre-Activation Computation

In Figure 9, we compare the kernel runtimes of the CUDA alternating and CUDA fused kernel with and without the external gate preactivation. `w/ Linear` denotes with external gate preactivation computation via a linear layer. The impact of the gate preactivation computation is marginal compared to the overall kernel runtime.

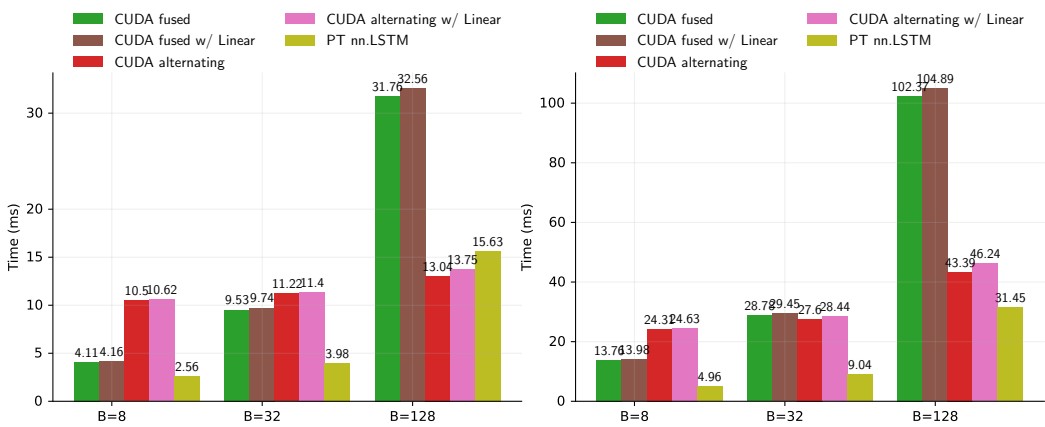

Figure 9: LSTM Runtime for different batch sizes (B) on a NVIDIA H100. We use one head with head dimension 768. We compare the kernel runtime with and without the gate preactivation matrix multiplication. **Left:** Forward pass. **Right:** Forward + backward pass.

## H.5 sLSTM Runtime Experiments

In Figures 10, 11, 12, 13 and 14, we show the results of the experiments from Section 6.1 for the sLSTM (Beck et al., 2024).

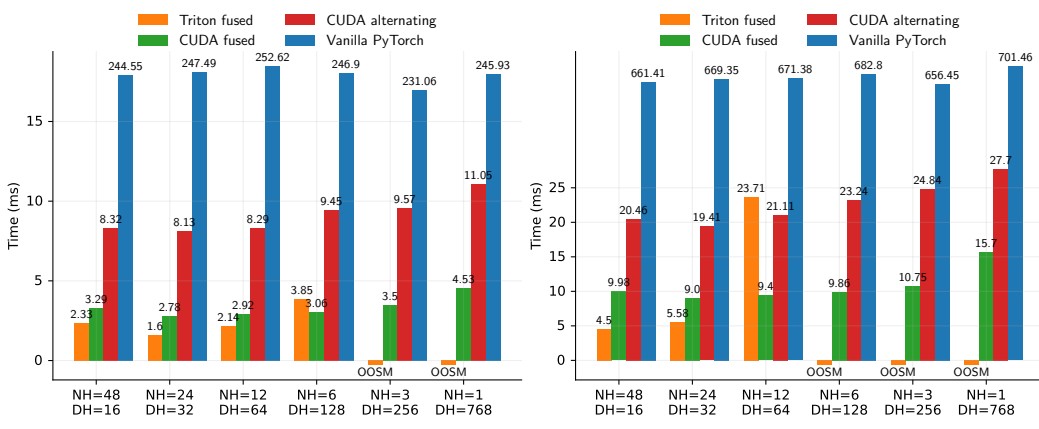

Figure 10: sLSTM Runtime for different head dimensions (DH) and number of heads (NH) on a NVIDIA H100. Overall embedding dimension is fixed at 768. We use batch size 16 and sequence length 1024. **Left:** Forward pass. **Right:** Forward + backward pass.

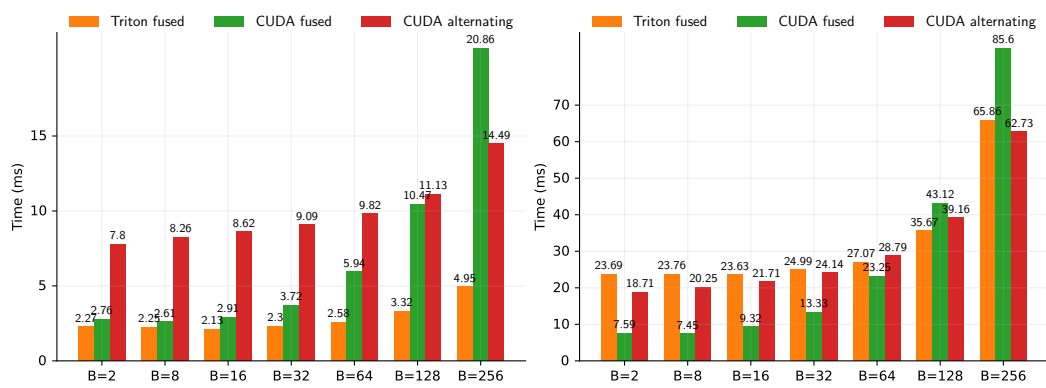

Figure 11: sLSTM Runtime for different batch sizes (B) on a NVIDIA H100, at 12 heads with head dimension 64 and sequence length 1024. **Left:** Forward pass. **Right:** Forward + backward pass.

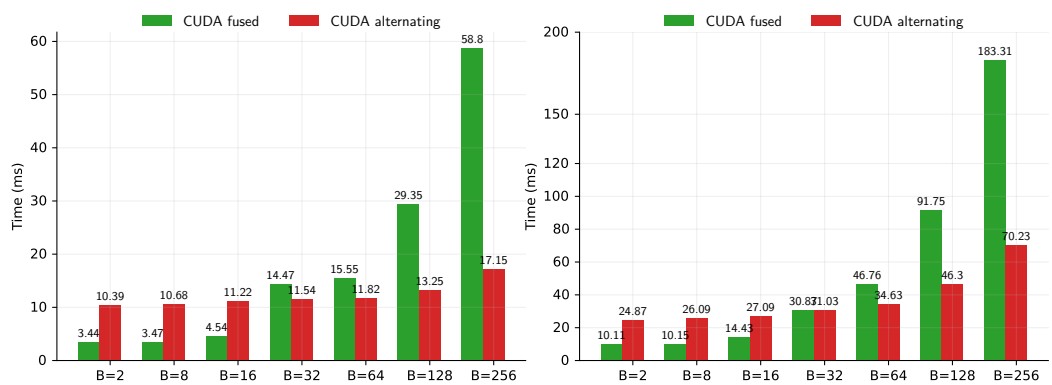

Figure 12: sLSTM Runtime for different batch sizes (B) on a NVIDIA H100, at one head with head dimension 768 and sequence length 1024. **Left:** Forward pass. **Right:** Forward + backward pass.

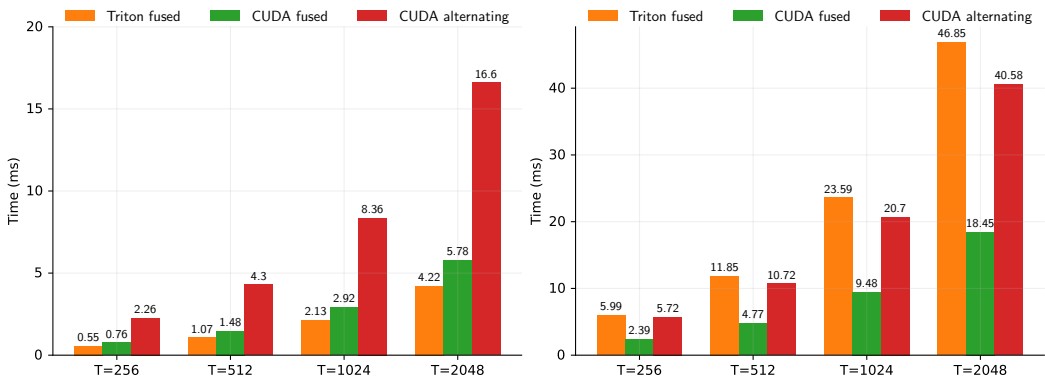

Figure 13: sLSTM Runtime for different sequence lengths (T) on a NVIDIA H100. We use 12 heads with head dimension 64 and batch size 16. **Left:** Forward pass. **Right:** Forward + backward pass.

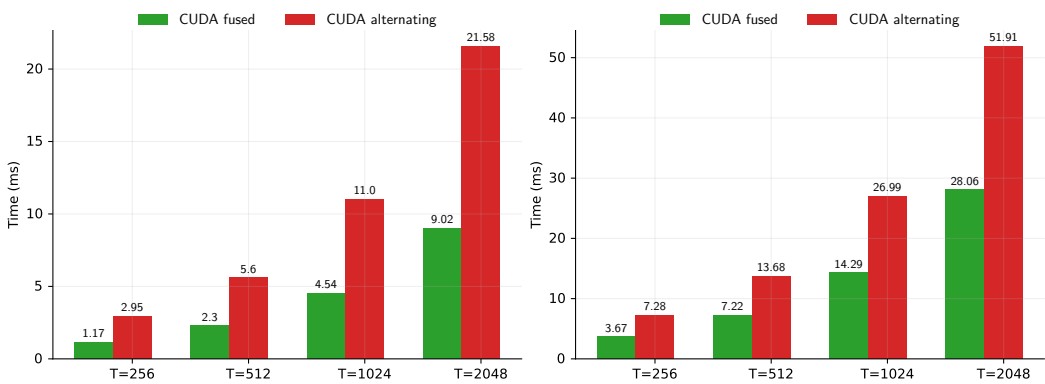

Figure 14: sLSTM Runtime for different sequence lengths (T) on a NVIDIA H100. We use one head with head dimension 768 and batch size 16. **Left:** Forward pass. **Right:** Forward + backward pass.

## H.6  NUMERICAL ERROR ANALYSIS

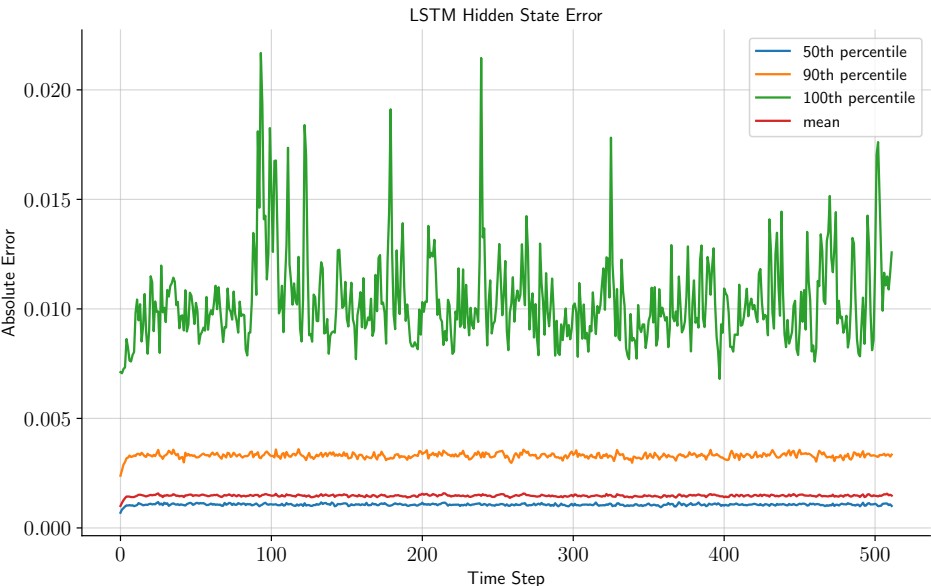

Figure 15: Numerical error of the CUDA fused kernel in bfloat16 compared to a vanilla PyTorch baseline in float64 over the sequence length. For an RNN, one would assume an accumulation of errors over multiple steps.

In Figure 15, we plot the numerical deviations in the LSTM hidden states (i.e. the outputs) over time. We compare our CUDA fused kernel in bfloat16 (the default setting) to our vanilla PyTorch implementation in float64. For this experiment we use a single example with sequence length 512 and a single head with head dimension 768. We use a random normal distribution to generate the weights, biases and inputs.

We plot the 50th, 90th and 100th percentiles of the absolute errors of the LSTM hidden state output-many per timestep. Percentiles are computed over the head dimension of 768.

There exist maximum deviations of about 0.01, but this error stabilizes over time.

## I  LANGUAGE MODEL TRAINING ON A100S

| Model | Heads | Param. (M) | Train Time (h) | Median Step (s) | Val PPL (val) |
|---|---|---|---|---|---|
| LSTM CUDA fused | 1 | 190 | 15.4 | 1.699 | 22.1 |
| LSTM CUDA altern. | 1 | 190 | 15.1 | 1.685 | 22.1 |
| LSTM PT nn.LSTM | 1 | 190 | 6.6 | 0.730 | 25.9 |
| sLSTM CUDA fused | 1 | 190 | 15.3 | 1.707 | 21.3 |
| sLSTM CUDA altern. | 1 | 190 | 15.6 | 1.720 | 21.3 |
| LSTM CUDA fused | 12 | 164 | 7.8 | 0.820 | 22.3 |
| LSTM CUDA altern. | 12 | 164 | 7.7 | 0.809 | 22.3 |
| sLSTM CUDA fused | 12 | 164 | 8.0 | 0.865 | 21.7 |
| sLSTM CUDA altern. | 12 | 164 | 8.0 | 0.852 | 21.7 |
| Transformer | 12 | 162 | 6.3 | 0.688 | 18.0 |

Table 3: 165M Model training on 15B tokens of SlimPajama on 8xA100s.

## J    LANGUAGE TRAINING DETAILS

All models are roughly at 165 M parameter scale, that means 12 Transformer blocks (post-up projection), with a swish-gated MLP and embedding dimension 768. The Transformer uses RoPE embeddings, whereas the other models do not use any additional positional information. We train with context length 1024 and a global batch size of 512, resulting in roughly 30 k training steps for 15 B tokens of the SlimPajama dataset. We use the GPT-2 tokenizer and learning rate 1e-3 with linear warmup over 750 steps and cosine decay to 1e-4 over 30k steps. We use PyTorch in version 2.4.0 and CUDA 12.1 for A100 and 12.4 for H100s. The training uses PyTorch FSDP in the NO_SHARD setting (DDP) with Automated Mixed Precision using bfloat16 and float32 for accumulations.
For the A100 experiments, we use one node of eight A100-SXM (80GB) GPUs and a local batch size of 64. For H100-SXM we reduce the local batch size to 32 and use 2 gradient accumulation steps due to `OutOfMemory` errors, even though they have the same HBM size (80 GB).
For the language model trainings, we see more spikes in the training step times for FlashRNN models compared to the PyTorch implementations, which should be investigated further.

## K    EXPERIMENTAL DETAILS PARITY TASK IN SEQUENCE EXTRAPOLATION

For the parity task we train on the parity task with varying training sequence lengths up to 40. For the reported validation, we evaluate on sequence lengths between 40 and 256. Sequence lengths are uniformly sampled in the respective ranges. We train on three seeds for learning rates {1e-2, 1e-3, 1e-4} and choose best learning rates. We train for up to 100k steps with batch size 256 with linear warmup of 10k and cosine decay to 10 % of the peak learning rate. Elman networks and LSTM cannot reach 100 % accuracy on sequence extrapolation for the smallest learning rate. All models reach low losses and high accuracies on the training set.

