# OpenReview forum: "FlashRNN: I/O-Aware Optimization of Traditional RNNs on modern hardware"
_ICLR.cc/2025/Conference — ICLR 2025 Poster_

### Official Review · Reviewer_z311 · 2024-10-24

**Soundness:** 3
**Presentation:** 2
**Contribution:** 2
**Rating:** 6
**Confidence:** 3

**Summary:**

FlashRNN is a hardware-optimized library that significantly accelerates traditional RNNs, like LSTMs and GRUs, using optimized kernels for GPUs. The paper introduces kernel-level optimizations that fuse matrix multiplications and activation functions, achieving up to 50x speed-ups compared to standard implementations. FlashRNN also supports parallel processing through a "multi-head" approach and uses the ConstrINT framework to model hardware constraints. This library enables faster and more efficient RNN performance, particularly in state-tracking tasks.

**Strengths:**

FlashRNN offers performance improvements by focusing on kernel fusion techniques that optimize LSTM and GRU operations on modern GPUs. The library’s focus on fusing matrix multiplication and activation functions into a single operation significantly reduces memory access overhead, a critical bottleneck in memory-bound computations. The integration of the ConstrINT framework for hardware-aware optimization ensures that the method adapts well to various GPU architectures. Additionally, the open-source nature of FlashRNN encourages further exploration and development in state-tracking tasks, where RNNs still excel. The paper’s effort to justify LSTM optimization versus Transformers provides valuable insight into the relevance of RNNs in certain domains.

**Weaknesses:**

FlashRNN has some limitations, particularly in its use cases, as its advantages are more pronounced for state-tracking tasks and less clear for applications where parallelizable models like Transformers excel, which may limit its broader appeal. Additionally, its focus on GPU-based optimizations reduces its applicability for users working with other hardware, such as TPUs or CPUs. Despite performance improvements, RNNs still face inherent scalability issues due to sequential processing, and FlashRNN doesn't fully address the parallelization advantage held by Transformers. Lastly, the paper notes numerical precision differences in training compared to implementations like PyTorch's cuDNN, which may need further investigation.

**Questions:**

A few questions and feedback:

Was the profiling conducted with torch.compile optimizations? Have you considered trying TVM auto-tuning for further optimization?

Since LSTM's arithmetic intensity is mostly memory-bound, the proposed method seems to focus primarily on fusion operations. It’s unclear if other techniques to tackle bandwidth bottlenecks for inference were explored. Given that Transformers were specifically designed to overcome LSTM limitations, this aspect could use more clarification.  The paper would benefit from a more detailed comparison of computational complexity scaling between FlashRNN and Transformers, particularly how each scales with respect to input size and hidden states.

While it’s good that the paper addresses why LSTM optimization remains relevant compared to Transformers, it is not fully convincing why FlashRNN would outperform FlashAttention or how FlashRNN differs fundamentally from applying FlashAttention-like methods to RNNs.

Has the method been tested on GPUs with less SRAM? It would be valuable to understand how FlashRNN performs in environments with different hardware constraints. What are the options if the weight matrix (W) cannot be cached? Addressing this scenario would help clarify how the method adapts to different hardware limitations.

---

> ### Author Response · Authors · 2024-11-20
>
> Dear z31124,
>
> thank you for your detailed review and all your questions and concerns. The speed comparison was conducted without torch.compile, as PyTorch would unroll the 1024 steps of the time loop, making it very slow to compile. In Appendix G.2, we added a comment on this, measuring the speed for short sequence lengths in exemplary tests on a RTX3090.
>
> This work’s focus is on training, where the whole sequence is given in advance. There might be further optimizations for inference going step by step.
>
> We have not explored the TVM compiler, because it is listed as „inference only“ backend in torch.compile() at the time of writing the paper and for our purposes we need Training&Inference backends. (See https://pytorch.org/docs/stable/torch.compiler.html).
> For torch.compile() we relied on the default backend „inductor“. Torch.compile() compiles the PyTorch code into Triton kernels. Since at the time of writing the paper torch.compile() was not able to compile the RNN in reasonable compile times, we decided to write our own custom Triton kernels for a comparison.
> We see that torch.compile is still in active development and hope that further generations will be more capable in compiling RNN workloads to efficient Triton kernels.
>
> While we see the recurrent matrix caching as the main speed contribution, there are other optimizations made internally to reach the measured speeds. This includes coalesced memory accesses, use of shared memory for intermediate results, shared memory padding to avoid memory bank conflicts, and using synchronizations only at the points and levels where needed to ensure numerical correctness. Since the RNN architectures optimized in FlashRNN are not sequence parallelizable in their standard computational structure, any sequence parallelizable architecture can out-speed this given enough hardware parallelization and sequence length. In Appendix Section G, we briefly compare the computational and memory complexity of RNNs to Transformer architectures now.
>
> FlashRNN does not intend to outperform FlashAttention in speed, nor in performance for tasks that do not need state tracking capabilities. Where these capabilities are needed however, it should serve as a library for speeding up research and investigating novel RNN architectures. While FlashRNN also falls into the category of kernel optimization, the approach here is very different to FlashAttention, as RNNs are not sequence parallelizable and typically have a large recurrent matrix that we are able to cache efficiently in our code.
> This is now stated as well in the benchmark introduction Section 6.1.
>
> In Appendix Section H.1, we added now the attainable head dimensions for our fused kernels on different GPUs - using the automatic adaption provided by ConstrINT. For larger head dimensions one has to switch to the alternating kernel, which supports arbitrary head dimensions at the cost of typically lower speed.

---

### Official Review · Reviewer_HW4x · 2024-11-01

**Soundness:** 3
**Presentation:** 3
**Contribution:** 3
**Rating:** 8
**Confidence:** 2

**Summary:**

The paper optimizes traditional RNNs on GPUs. They introduce optimized kernels and an optimization framework. One challenge for efficient and scalable implementations that the authors tackle is the requirement of strict sequential processing of RNNs. They introduce a parallelization variant that processes multiple RNNs of smaller hidden state in parallel, inspired by the head-wise processing in transformers. They implement two variants, an alternating version switching between point-wise and matrix-multiplication kernels and a fused implementation, optimizing memory transfers while using hardware-optimized matrix-multiplication. The second leads to a further 3-4x speed-up over the alternating option for small batch sizes. Their kernels can achieve 50x speed-ups over a vanilla PyTorch implementation and allow 40x larger hidden sizes compared to their Triton implementation.
To enable flexibility on different GPU variants, they also introduce an optimization framework for hardware-internal cache sizes, memory and compute handling.

**Strengths:**

- The paper is overall well-written, uses clear language and structure.
- it tackles an important problem of overcoming a fundamental limitation important for further scaling rnns. This would enable state-intensive tasks where state continuity and stepwise dependency matter, which might be less naturally handled by Transformers
- They compare to relevant Backends (e.g., cuDNN)

**Weaknesses:**

They mention the HASTE library and that they overcome their limitations. However, I think they don't benchmark directly against HASTE which would be relevant.

**Questions:**

Can you elaborate on the similarities and differences of the HASTE implementation and your approach. Did you benchmark against HASTE? If not, why not?

---

> ### Author Response · Authors · 2024-11-20
>
> Dear vHW4x01,
>
> thanks for your nice review!
> Haste is very similar to the alternating version of our kernels, which is why we did not do a speed comparison in the first place. We have added Haste now to our speed measurements. It is slower than our alternating kernels as it is limited to float32 and float64 precision. Since haste has not been updated for around four years and it does not support float16 and bfloat16 precision. These precisions are integrated in FlashRNN and strongly benefit from TensorCore acceleration and memory speedups. Feel free to look also at the improvements, additions and clarifications made in response to the other reviewers, and to add further questions.

---

> > ### Comment · Reviewer_HW4x · 2024-11-25
> >
> > Thank you for the addition of Haste and the provided context. I will keep my score.

---

### Official Review · Reviewer_fH1r · 2024-11-04

**Soundness:** 3
**Presentation:** 2
**Contribution:** 2
**Rating:** 6
**Confidence:** 3

**Summary:**

While Transformers are state-of-the-art in sequence modeling, they lack state-tracking abilities. RNN families, like LSTMs and GRUs, can perform state tracking but at the expense of sequential processing. To address this, this paper introduces FlashRNN, which optimizes kernel operations to leverage GPU registers and SRAM. Additionally, it incorporates ConstrINT, an auto-tuning optimization tool that operates under hardware constraints, such as cache and memory limits, by solving an integer constraint satisfaction problem. Experimental results demonstrate the effectiveness and efficiency of FlashRNN.

**Strengths:**

+ Hardware-aware optimization for accelerating RNN model training and inference is promising.
+ It claims that FlashRNN will be open-sourced, which helps further research in this direction.

**Weaknesses:**

- Certain critical procedures are somewhat unclear and would benefit from further clarification.
- The experimental section could be strengthened with additional details and improvements.

**Questions:**

1. The interaction between ConstrINT and other components in FlashRNN is somewhat unclear. Providing an overall architecture or workflow for FlashRNN would be helpful.
2. Algorithm 5.1 outlines the kernel fusion procedure but lacks a detailed explanation. A further illustration of this algorithm would be beneficial.
3. In Algorithm 5.1, line 242 mentions “Load RtbgsR_{tbgs}Rtbgs​, BtgB_{tg}Btg​ to registers and SRAM.” How many registers and how much SRAM can be used for storing these values? Utilizing all available registers and SRAM may not be ideal, as they may also be needed for storing intermediate results during computation. If this strategy is used, it could impact performance. Detailed clarification is needed.
4. Line 299 states that multiple heads are computed in parallel in different programs without synchronization. Could this lack of synchronization impact accuracy?
5. Lines 301 to 303 indicate that custom caching strategies cannot be applied due to restrictions on GPU register access in Triton. Is there a significant gap between the intended design in Algorithm 5.1 and the Triton implementation? How large is this gap?
6. Line 310 mentions that FlashRNN pads with zeros to support small batch sizes, at the cost of efficiency. How much efficiency is lost due to this padding?
7. The evaluation setup is somewhat confusing. Are all four backends (CUDA fused, CUDA alternating, Triton fused, and Vanilla PyTorch) being compared with the two baselines, FlashAttention2 and nn.LSTM? If so, it appears that FlashRNN’s performance is lower than the baselines in most configurations shown in Figures 1 to 4. Additionally, why is FlashRNN compared with FlashAttention2?
8. In the experiments, lines 467 to 468 mention replacing attention blocks with FlashRNN LSTM for speed comparisons. Does this substitution provide a meaningful comparison?
9. None of the experiments include accuracy comparisons, which appears to be a limitation.

---

> ### Author Response · Authors · 2024-11-20
>
> Dear fH1r04, thank you for your review and comments. Thanks for your detailed questions that we would like to address:
> 1. In the new revision of our paper, we added Appendix Section D to describe the higher level interaction of ConstrINT with the kernel compilation process.
> 2. Also, we provide a more detailed outline of the central algorithm in Appendix Section B.
> 3. Regarding the shared memory and registers used, we follow the strategy to use as much shared memory and registers as possible for the cached values (mostly the recurrent weight matrix). This was beneficial for the larger head dimensions of 768 and 1024, as otherwise the recurrent matrix would not fit at all. A more detailed tuning can be done by adapting the register use manually in the ConstrINT variables / kernel parameters.
> 4. The lack of synchronization does not impact accuracy in a numerical sense. One can view the different heads also as smaller recurrent neural networks that run in parallel, without any approximations.
> While there is no synchronization of the different heads at each time step, there is a synchronization of all outputs after computing the whole recurrent sequence. This is needed for a subsequent layer that potentially uses them in time-parallel fashion (e.g. in an MLP).
> 5. That is true. In Triton we do not have access to GPU registers, therefore we cannot keep the recurrent weights and the biases in registers manually. Instead we rely on the Triton compiler to manage SRAM and Register transfers.
> There is no significant gap in the design of the fused FlashRNN algorithm between CUDA and Triton. Both support the headwise parallelization and avoid multiple loading the recurrent weights and biases multiple times.
> However, as CUDA gives more fine grained control over GPU Hardware features such as the mentioned register level access and grid synchronization for example.
> Therefore, in our CUDA implementation we make use of these features to further optimize the kernels. Register level caching of the recurrent weights for example enables to avoid unnecessary loads from SRAM to registers in every time step.
> Grid synchronization allows to distribute one head to multiple thread blocks and accumulate the R s matrix multiply results over HBM. This enables the much larger head dimensions (up to 2688) in contrast to the limited size of 64 or 128 in Triton.
> In our revised paper we have added an Appendix E and Algorithm 5 which provide further details on the Triton implementation.
> 6. The minimal batch size is 8 as this is the minimal size of a TensorCore matrix multiplication in one outer dimension. This means that only 1/8th of the FLOPs are used for batch size one. However due to the order of magnitude higher FLOPs-speed of tensor cores this should overcompensate the efficiency drop. While the alternating kernel in Figure 4 does not have the limitation of the batch size, and may not suffer from the efficiency drop, it is still slower compared to the fused one.
> 7. We clarified our setup in Section 6.1 . The FlashAttention 2 baseline serves as a speed baseline of a common sequence modeling layer, which however does not have state tracking capabilities. The nn.LSTM baseline for the given hidden dimension is limited to one head (there is no multi-head version), and based on the cuDNN implementation of LSTM that is closed source. It is therefore not possible to try out new RNN architectures with nn.LSTM, but it can serve as a baseline of an industrially optimized kernel.
> 8. This is a setting Attention is commonly used in (Transformer), and RNN models can be used in as well. In the xLSTM paper, the sLSTM-only models were used in similar blocks. Our experiments show that training is feasible and not too much slower than attention based training. As previously mentioned, RNNs are currently the only hardware optimized sequence models with state tracking capabilities. Tasks that need those will therefore benefit from our kernels.
> 9. The tests in the code check our models for differences to a PyTorch baseline, where we see differences due to the low precision. We follow good practices, as in our kernels we use float32 accumulation internally, which potentially is not the case for the nn.LSTM baseline. This might be the cause for the difference in language model training performance between the FlashRNN and the nn.LSTM baseline. In addition, in Appendix H.6 we plot the numerical error of our CUDA fused kernel in bfloat16 compared to a vanilla PyTorch implementation in float64 over time. We see that the maximum errors stabilize in a modest range of 0.01 over time.
>
>
> Thanks for all your detailed questions! We hope we could improve upon your previous impression of our paper, address most of your concerns and clarify the unclear points. We are happy to take further comments and suggestions!

---

### Official Review · Reviewer_8hLb · 2024-11-04

**Soundness:** 2
**Presentation:** 3
**Contribution:** 2
**Rating:** 6
**Confidence:** 2

**Summary:**

This paper introduces FlashRNN, which optimizes traditional RNNs on modern GPUs. It divides RNNs into smaller pieces and parallelize over it to increase efficiency on GPUs. The model auto-adjusts its internal cache settings using a constraint satisfaction problem solver, enabling further performance optimization based on hardware. The experimental results show that the proposed solution is 50x faster than a vanilla PyTorch implementation.

**Strengths:**

- The paper is easy to follow and well organized.
- Hardware aware fine tuning
- Comprehensive benchmarking

**Weaknesses:**

- The paper emphasizes the register level CUDA optimization. It'd be great if the authors can show how the implementation is close to the roofline through profiling.
- I believe comparing to pytorch is not fair as pytorch has overhead to launch its internal CUDA kernel.
- Several fusion can be achieved through compilation. How is it different? Also, not using compilation seems to be unfair comparison.
- Why FlashAttention2 matters in the RNN?

**Questions:**

Please see the weakness section above.

---

> ### Author Response · Authors · 2024-11-20
>
> Dear 8hLb,
>
> thank you for your review and comments on our paper. In the Appendix Section F  we added the requested roofline plots using NVIDIA Nsight Compute Utils (ncu). There, one can see that the fused kernels have a higher arithmetic intensity compared to the alternating ones. The forward kernel also comes close to the closed-source cuDNN baseline, but there is still a gap the hardware limitations of the roofline.
>
> We agree that comparing to pure PyTorch is unfair, but this is the typical entry point for researchers. We have tried torch.compile to speed up the vanilla PyTorch version, but this unrolls the whole recurrent computation graph along the sequence dimension, since there is no generic scan in PyTorch currently. This leads to excessively long compilation times, especially for deeper architectures.
>
> As mentioned in Section 6.1 in the new revision of our paper, we compare to FlashAttention2 as a very common sequence modeling kernel. Attention however, does not have state tracking capabilities. It just serves as a baseline in terms of speed, when these capabilities are not needed.
>
> We hope that our additional experiments and explanations could improve your impression over previous concerns. We are also happy to take in further comments or suggestions for improvement.

---

> > ### Comment · Reviewer_8hLb · 2024-11-24
> >
> > Thanks for the explanation and clarification! That resolves most of my concerns. I just raised the score.

---

### Author Response · Authors · 2024-11-20

We thank all the reviewers for their constructive comments and valuable feedback.

We are glad that the reviewers liked our work on hardware optimization for RNN architectures with state tracking capabilities.

We want to highlight the following major additions in the paper:
- Integration of the haste library [1] as an additional baseline
- Appendix Section B: FlashRNN Fused Algorithm in more detail
- Appendix Section D: ConstrINT Kernel Solution and Optimization Process
- Appendix Section E: Triton Algorithm
- Appendix Section F: Roofline Analysis
- Appendix Section G: Computational Complexity of RNNs compared to Attention
- Appendix Section H.1: Head Dimension Limits for different Hardware
- Appendix Section H.2: Note on torch.compile Experiments
- Appendix Section H.6: Numerical Error Analysis

We want to emphasize again that FlashRNN does not intend to compete against FlashAttention in terms of speed or against Transformers in Language modeling in its current form. However, Attention and current competitors do not have state tracking capabilities [2] that traditional RNNs do have, which is why we want to optimize these for speed on current GPU hardware to check if this is an actual limitation. We can show that, while still being slower than FlashAttention and Transformers, with our optimizations one can reach reasonable speeds that do neither limit experimentation with new RNN architectures nor moderate scaling to 165 M parameter sized models - reaching a ~50x speed up over vanilla PyTorch.

[1] S. Nanavati 2020: Haste: a fast, simple, and open RNN library

[2] W. Merrill et al. 2024: The Illusion of State in State-Space Models

---

### Meta-Review · Area_Chair_eWuz · 2024-12-11

**Metareview:**

The paper proposes several techniques to improve the performance of RNN, such as adopting the multi-head architecture of Transformers in RNNs, optimized kernels for faster activation and backpropagation computation, and adaptation of said kernels for different hardware cache sizes.

Reviewers generally agreed the paper's major strengths are closing the implementation gap between RNNs and Transformer model (which can facilitate more experimentation and research into alternative model architectures), and the hardware-aware kernel library that can adapt to hardware of different cache sizes. At the same time, reviewers noted that the gap between RNNs and Transformers is not fully closed by this paper alone. Because it is not easy to improve the performance of RNNs (which has merit as a well-studied architecture) against the much larger volume of ongoing Transformer implementation optimization research, it is fair to say that multiple works in RNN optimization would be needed to change the state of the art, and that this paper makes a solid contribution towards that goal.

**Additional Comments On Reviewer Discussion:**

Reviewers had concerns about technical clarity and experiment setup, which were addressed through discussion with the authors. The broader issue of RNNs still not being on par with Transformers was discussed in the AC discussion, and reviewers agreed that the paper should be accepted in spite of this limitation.

---

### Decision · Program_Chairs · 2025-01-22

Accept (Poster)